# Residual Loss Prediction: Reinforcement Learning with no Incremental Feedback

**Hal Daumé III** [*]
University of Maryland &
Microsoft Research NYC
me@hal3.name

**John Langford**
Microsoft Research NYC
jcl@microsoft.com

**Amr Sharaf**
University of Maryland
amr@cs.umd.edu

## Abstract

We consider reinforcement learning and bandit structured prediction problems with very sparse loss feedback: only at the end of an episode. We introduce a novel algorithm, Residual Loss Prediction (Reslope), that solves such problems by automatically learning an internal representation of a denser reward function. Reslope operates as a reduction to contextual bandits, using its learned loss representation to solve the credit assignment problem, and a contextual bandit oracle to trade-off exploration and exploitation. Reslope enjoys a no-regret reduction-style theoretical guarantee and outperforms state of the art reinforcement learning algorithms in both MDP environments and bandit structured prediction settings.

## 1 Introduction

Current state of the art learning-based systems require enormous, costly datasets on which to train supervised models. To progress beyond this requirement, we need learning systems that can interact with their environments, collect feedback (a loss or reward), and improve continually over time. In most real-world settings, such feedback is sparse and delayed: most decisions made by the system will not immediately lead to feedback. Any sort of interactive system like this will face at least two challenges: the credit assignment problem (which decision(s) did the system make that led to the good/bad feedback?); and the exploration/exploitation problem (in order to learn, the system must try new things, but these could be bad).

We consider the question of how to learn in an extremely sparse feedback setting: the environment operates episodically, and the only feedback comes at the *end* of the episode, with *no incremental feedback* to guide learning. This setting naturally arises in many classic reinforcement learning problems (§4): a barista robot will only get feedback from a customer after their cappuccino is finished[1]. It also arises in the context of bandit structured prediction (Sokolov et al., 2016; Chang et al., 2015) (§2.2), where a structured prediction system must produce a single output (e.g., translation) and observes only a scalar loss.

We introduce a novel reinforcement learning algorithm, Residual Loss Prediction (Reslope) (§3), which aims to learn *effective representations of the **loss** signal*. By effective we mean effective in terms of credit assignment. Intuitively, Reslope attempts to learn a decomposition of the episodic loss into a sum of per-time-step losses. This process is akin to how a person solving a task might realize before the task is complete when and where they are likely to have made suboptimal choices. In Reslope, the per-step loss estimates are conditioned on all the information available up to the current point in time, allowing it to learn a highly non-linear representation for the episodic loss (assuming the policy class is sufficiently complex; in practice, we use recurrent neural network policies). When the system receives the final episodic loss, it uses the *difference* between the observed loss and the cumulative predicted loss to update its parameters.

---

[*] Authors are listed alphabetically.

[1] This problem can be—and to a large degree *has* been—mitigated through the task-specific and complex process of reward engineering and reward shaping. Indeed, we were surprised to find that many classic RL algorithms fail badly when incremental rewards disappear. We aim to make such problems disappear.

Algorithmically, RESLOPE operates as a *reduction* (§3.3) to contextual bandits (Langford & Zhang, 2008), allowing the bandit algorithm to handle exploration/exploitation and focusing only on the credit assignment problem. RESIDUAL LOSS PREDICTION is theoretically motivated by the need for variance reduction techniques when estimating counterfactual costs (Dudík et al., 2014) and enjoys a no-regret bound (§3.3) when the underlying bandit algorithm is no-regret. Experimentally, we show the efficacy of RESLOPE on four benchmark reinforcement problems and three bandit structured prediction problems (§5.1), comparing to several reinforcement learning algorithms: Reinforce, Proximal Policy Optimization and Advantage Actor-Critic.

## 2 PROBLEM FORMULATION AND BACKGROUND

We focus on finite horizon, episodic Markov Decision Processes (MDPs) in this paper, which captures *both* traditional reinforcement learning problems (§4) *and* bandit structured prediction problems (§2.2). Our solution to this problem, RESIDUAL LOSS PREDICTION (§3) operates in a *reduction* framework. Specifically, we assume there exists "some" machine learning problem that we know how to solve, and can treat as an oracle. Our reduction goal is to develop a procedure that takes the reinforcement learning problem described above and map it to this oracle, so that a good solution to the oracle guarantees a good solution to our problem. The specific oracle problem we consider is a contextual bandit learning algorithm, relevant details of which we review in §2.1.

Formally, we consider a (possibly virtual) learning agent that interacts directly with its environment. The interaction between the agent and the environment is governed by a restricted class of finite-horizon Markov Decision Processes (MDP), defined as a tuple $\{\mathcal{S}, s_0, \mathcal{A}, \mathcal{P}, \mathcal{L}, H\}$ where: $\mathcal{S}$ is a large but finite state space, typically $\mathcal{S} \subset \mathbb{R}^d$; $s_0 \in \mathcal{S}$ is a start state; $\mathcal{A}$ is a finite action space[2] of size $K$; $\mathcal{P} = \{\mathcal{P}(s'|s, a) : s, s' \in \mathcal{S}, a \in \mathcal{A}\}$ is the set of Markovian transition probabilities; $\mathcal{L} \in \mathbb{R}^{|\mathcal{S}|}$ is the state dependent loss function, defined only at terminal states $s \in \mathcal{S}$; $H$ is the horizon (maximum length of an episode).

The goal is to learn a policy $\pi$, which defines the behavior of the agent in the environment. We consider policies that are potentially functions of entire trajectories[3], and potentially produce distributions over actions: $\pi(s) \in \Delta^{\mathcal{A}}$, where $\Delta^{\mathcal{A}}$ is the $\mathcal{A}$-dimensional probability simplex. However, to ease exposition, we will present the background in terms of policies that depend only on states; this can be accomplished by simply blowing up the state space.

Let $d_h^\pi$ denote the distribution of states visited at time step $h$ when starting at state $s_0$ and operating according to $\pi$: $d_{h+1}^\pi(s') = \mathbb{E}_{s_h \sim d_h^\pi, a_h \sim \pi(s_h)} \mathcal{P}(s' \mid s = s_h, a = a_h)$ The quality of the policy $\pi$ is quantified by its value function or q-value function: $V^\pi(s) \in \mathbb{R}$ associates each state with the expected future loss for starting at this state and following $\pi$ afterwards; $Q^\pi(s, a) \in \mathbb{R}$ associates each state/action pair with the same expected future loss: $V^\pi(s_h) = \mathbb{E}_{s_H \sim d_H^\pi \mid s_h} \mathcal{L}(s_H)$ and $Q^\pi(s_h, a_h) = \mathbb{E}_{s_H \sim d_H^\pi \mid s_h, a_h} \mathcal{L}(s_H)$ The learning goal is to estimate a policy $\pi$ from a hypothesis class of policies $\Pi$ with minimal expected loss: $J(\pi) = V^\pi(s_0)$.

### 2.1 CONTEXTUAL BANDITS

The contextual bandit learning problem (Langford & Zhang, 2008) can be seen as a tractable special case of reinforcement learning in which the time horizon $H = 1$. In particular, the world operates episodically. At each round $t$, the world reveals a context (i.e. feature vector) $\boldsymbol{x}_t \in \mathcal{X}$; the system chooses an action $a_t$; the world reveals a scalar loss $\ell_t(\boldsymbol{x}_t, a_t) \in \mathbb{R}^+$, a loss *only* for the selected action that may depend stochastically on $\boldsymbol{x}_t$ and $a_t$. The total loss for a system over $T$ rounds is the sum of losses: $\sum_{t=1}^{T} \ell_t(\boldsymbol{x}_t, a_t)$. The goal in policy optimization is to learn a policy $\pi : \boldsymbol{x} \to \mathcal{A}$ from a policy class $\Pi$ that has low *regret* with respect to the best policy in this class. Assuming the learning algorithm produces a sequence of policies $\pi_1, \pi_2, \ldots, \pi_T$, its regret is: $Regret\left(\langle \pi_t \rangle_{t=1}^T\right) = \sum_{t=1}^T \ell(\boldsymbol{x}_t, \pi_t(\boldsymbol{x}_t)) - \min_{\pi^* \in \Pi} \sum_{t=1}^T \ell(\boldsymbol{x}_t, \pi^*(\boldsymbol{x}_t))$. The particular contextual bandit algorithms we will use in this paper perform a second level of reduction: they assume access to an oracle supervised learning algorithm that can optimize a cost-sensitive loss (Appendix C), and transform

---

[2]In some problems the set of actions available will depend on the current state.

[3]Policies could choose to ignore all but the most recent state, for instance in fully observable environments, though this may be insufficient in partially observable environments (Littman & Sutton, 2002).

the contextual bandit problem to a cost-sensitive classification problem. Algorithms in this family typically vary along two axes: how to explore (faced with a new $x$ how does the algorithm choose which action to take); and how to update (Given the observed loss $\ell_t$, how does the algorithm construct a supervised training example on which to train). More details are in Appendix A.

## 2.2 BANDIT STRUCTURED PREDICTION VIA LEARNING TO SEARCH

In structured prediction, we observe structured input sequences $x^{\text{SP}} \in \mathcal{X}$ and the goal is to predict a set of correlated output variables $y^{\text{SP}} \in \mathcal{Y}$. For example, in machine translation, the input $x^{\text{SP}}$ is a sentence in an input language (e.g., Tagalog) and the output $y^{\text{SP}}$ is a sentence in an output language (e.g., Chippewa). In the fully supervised setting, we have access to samples $(x^{\text{SP}}, y^{\text{SP}})$ from some distribution $\mathcal{D}$ over input/output pairs. Structured prediction problems typically come paired with a structured loss $\ell(y^{\text{SP}}, \hat{y}^{\text{SP}}) \in \mathbb{R}^+$ that measures the fidelity of a predicted output $\hat{y}^{\text{SP}}$ to the "true" output $y^{\text{SP}}$. The goal is to learn a function $f : \mathcal{X} \to \mathcal{Y}$ with low expected loss under $\mathcal{D}$: $\mathbb{E}_{(x^{\text{SP}}, y^{\text{SP}}) \sim \mathcal{D}} \ell(y^{\text{SP}}, f(x^{\text{SP}}))$. Recently, it has become popular to solve structured prediction problems incrementally using some form of recurrent neural network (RNN) model. When the output $y^{\text{SP}}$ contains multiple parts (e.g., words in a translation), the RNN can predict each word in sequence, conditioning each prediction on all previous decisions. Although typically such models are trained to maximize cross-entropy with the gold standard output (in a fully supervised setting), there is mounting evidence that this has similar drawbacks to pre-RNN techniques, such as overfitting to gold standard prefixes (the model never learns what to do once it has made an error) and sensitivity to errors of different severity (due to error compounding). In order to achieve this we must formally map from the structured prediction problem to the MDP setting; this mapping is natural and described in detail in Appendix B.

Our focus in this paper is on the recently proposed *bandit* structured prediction setting (Chang et al., 2015; Sokolov et al., 2016), at training time, we only have access to input $x^{\text{SP}}$ from the marginal distribution $\mathcal{D}^{\mathcal{X}}$. For example, a Chippewa speaker sees an article in Tagalog, and asks for a translation. A system then produces a *single* translation $\hat{y}^{\text{SP}}$, on which a single "bandit" loss $\ell(\hat{y}^{\text{SP}} \mid x^{\text{SP}})$ is observed. Given only this bandit feedback, without ever seeing the "true" translation, the system must learn.

## 3 PROPOSED APPROACH

Our goal is to learn a good policy in a Markov Decision Process (§2) in which losses only arrive at the end of episodes. Our solution, RESIDUAL LOSS PREDICTION (RESLOPE), automatically deduces per-step losses based only on the episodic loss. To gain an intuition for how this works, suppose you are at work and want to meet a colleague at a nearby coffee shop. In hopes of finding a more efficient path to the coffee shop, you take a different path than usual. While you're on the way, you run into a friend and talk to them for a few minutes. You then arrive at the coffee shop and your colleague tells you that you are ten minutes late. To estimate the value of the different path, you wonder: how much of this ten minutes is due to taking the different path vs talking to a friend. If you can accurately estimate that you spent seven minutes talking to your friend (you lost track of time), you can conclude that the disadvantage for the different path is three minutes.

RESLOPE addresses the problem of **sparse reward signals** and **credit assignment** by learning a decomposition of the reward signal, essentially doing automatic reward shaping (evaluated in §5.3). Finally, it addresses the problem of **exploration vs exploitation** by relying on a strong underlying contextual bandit learning algorithm with provably good exploration behavior.

### 3.1 KEY IDEA: RESIDUAL LOSS PREDICTION

Akin to the coffee shop example, RESLOPE learns a decomposition of the episodic loss (i.e total time spent from work to the coffee shop) into a sum of per-time-step losses (i.e. timing activities along the route). RESLOPE operates as a reduction from reinforcement learning with episodic loss to contextual bandits. In this way, RESLOPE solves the *credit assignment* problem by predicting residual losses, and relies on the underlying contextual bandit oracle to solve explore/exploit. RESLOPE operates online, incrementally updating a policy $\pi^{\text{learn}}$ once per episode. In the structured

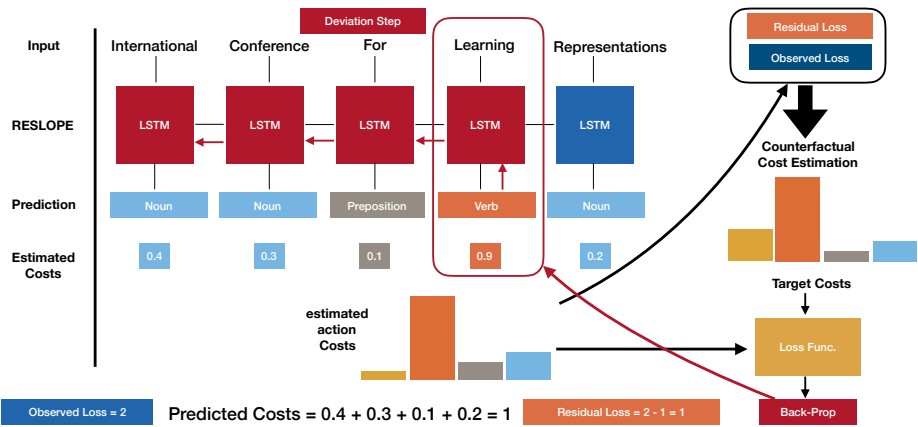

Figure 1: RESIDUAL LOSS PREDICTION: the system assigns a part-of-speech tag sequence to the sentence "International Conference for Learning Representations". Each state represents a partial labeling. The end state $e = $ [Noun, Noun, Preposition, Verb, Noun]. The end state $e$ is associated with an episodic loss $\ell(e)$, which is the total hamming loss in comparison to the optimal output structure $e^* = $ [Adjective, Noun, Preposition, Noun, Noun]. We emphasize that our algorithm doesn't assume access to neither the optimal output structure, nor the hamming loss for every time step. Only the total hamming loss is observed in this case ($\ell(e) = 2$).

contextual bandit setting, we assume access to a *reference policy*, $\pi^{\text{ref}}$, that was perhaps pretrained on supervised data, and which we wish to improve; a hyperparameter $\beta$ controls how much we trust $\pi^{\text{ref}}$. As $\pi^{\text{learn}}$ improves, we replace $\pi^{\text{ref}}$ with $\pi^{\text{learn}}$. In the RL setting, we set $\beta = 0$.

We initially present a simplified variant of RESLOPE that mostly follows the learned policy (and the reference policy as appropriate), except for a *single* deviation per episode. This algorithm closely follows the bandit version of the Locally Optimal Learning to Search (LOLS) approach of Chang et al. (2015), with three key differences: (1) residual loss prediction; (2) alternative exploration strategies; (3) alternative parameter update strategies. We assume access to a contextual bandit oracle CB that supports the following API:

1. CB.ACT($\pi^{\text{learn}}, \boldsymbol{x}$), where $\boldsymbol{x}$ is the input example; this returns a tuple $(a, p)$, where $a$ is the selected action, and $p$ is the probability with which that action was selected.

2. CB.COST($\pi^{\text{learn}}, \boldsymbol{x}, a$) returns the estimated cost of taking action $a$ in the context.

3. CB.UPDATE($\pi^{\text{learn}}, \boldsymbol{x}, a, p, c$), where $\boldsymbol{x}$ is the input example, $a \in [K]$ is the selected action, $p \in (0, 1]$ is the probability of that action, and $c \in \mathbb{R}$ is the target cost.

The requirement that the contextual bandit algorithm also predicts costs (CB.COST) is somewhat non-standard, but is satisfied by many contextual bandit algorithms in practice, which often operate by regressing on costs and picking the minimal predicted cost action. We describe the specific contextual bandit approaches we use in §3.2.

Algorithm 1 shows how our reduction is constructed formally. It uses a MAKEENVIRONMENT($t$) function to construct a new environment (randomly in RL and by selecting the $t$th example in bandit structured prediction). To learn a good policy, RESLOPE reduces long horizon trajectories to single-step contextual bandit training examples. In each episode, RESLOPE picks a single time step to deviate. Prior to the deviation step, it executes $\pi^{\text{learn}}$ as a roll-in policy and after the deviation step, it executes a $\beta$ mixture of $\pi^{\text{learn}}$ and $\pi^{\text{ref}}$ (Figure 5). *At* the deviation step, it calls CB.ACT to handle the exploration and choose an action. At *every* step, it calls CB.COST to estimate the cost of that action. Finally, it constructs a single contextual bandit training example for the deviation step, whose input was the observation at that step, whose action and probability are those that were selected by CB.ACT, and whose cost is the observed total cost *minus* the cost of every *other* action taken in this trajectory. This example is sent to CB.UPDATE. When the contextual bandit policy is an RNN (as in our setting), this will then compute a loss which is back-propagated through the RNN.

---

**Algorithm 1** RESIDUAL LOSS PREDICTION (RESLOPE) with *single* deviations

---

**Require:** Reference policy $\pi^{\text{ref}}$, mixture parameter $\beta \in [0,1]$, contextual bandit oracle CB, MAKEENVIRONMENT to build new enviornments
  1: Initialize a policy $\pi_0^{\text{learn}}$ {either randomly or from a pretrained model}
  2: **for all** episodes $t = 1 \ldots T$ **do**
  3:     env $\leftarrow$ MAKEENVIRONMENT($t$)
  4:     Initialize variables: example $\boldsymbol{x}^{\text{dev}}$, action $a^{\text{dev}}$, probability $p^{\text{dev}}$
  5:     Initialize cost vector $\hat{c}_h^{\text{dev}} = 0$ for $h = 1 \ldots \text{env}.H$
  6:     Choose deviation step $h^{\text{dev}} \leftarrow$ UNIFORM(env.$H$)
  7:     Choose rollout policy $\pi^{\text{mix}}$ to be $\pi^{\text{ref}}$ with probability $\beta$ or $\pi_{t-1}^{\text{learn}}$ with probability $1 - \beta$
  8:     **for all** time steps $h = 1 \ldots \text{env}.H$ **do**
  9:       $\boldsymbol{x} \leftarrow$ env.STATEFEATURES {computed by an RNN}
10:       **if** $h \neq h^{\text{dev}}$ { no deviation } **then**
11:         $a \leftarrow \begin{cases} \pi_{t-1}^{\text{learn}}(\boldsymbol{x}) & \text{if } h < h^{\text{dev}} \\ \pi^{\text{mix}}(\boldsymbol{x}) & \text{if } h > h^{\text{dev}} \end{cases}$
12:       **else if** $h = h^{\text{dev}}$ { deviation } **then**
13:         $(a^{\text{dev}}, p^{\text{dev}}) \leftarrow$ CB.ACT($\pi^{\text{learn}}, \boldsymbol{x}$)
14:         $\boldsymbol{x}^{\text{dev}} \leftarrow \boldsymbol{x}$
15:         $a \leftarrow a^{\text{dev}}$
16:       **end if**
17:       $\hat{c}_h^{\text{dev}} \leftarrow$ CB.COST($\pi_{t-1}^{\text{learn}}, \boldsymbol{x}, a$)
18:       env.STEP(a) {updates environment and internal state of the RNN }
19:     **end for**
20:     $\ell^{\text{residual}} \leftarrow$ env.FINALLOSS $- \sum_{h \neq h^{\text{dev}}} \hat{c}_h^{\text{dev}}$
21:     $\pi_t^{\text{learn}} \leftarrow$ CB.UPDATE($\pi_{t-1}^{\text{learn}}, \boldsymbol{x}^{\text{dev}}, a^{\text{dev}}, p^{\text{dev}}, \ell^{\text{residual}}$)
22: **end for**
23: Return average policy $\bar{\pi} = \frac{1}{T} \sum_t \pi_t^{\text{learn}}$

---

## 3.2 CONTEXTUAL BANDIT ORACLE

The contextual bandit oracle receives examples where the cost for only one predicted action is observed, but no others. It learns a policy for predicting actions minimizing expected loss by estimating the unobserved target costs for the unpredicted actions and exploring different actions to balance the exploitation exploration trade-off ($\S 3.2$). The contextual bandit oracle then calls a cost-sensitive multi-class oracle (Appendix C) given the target costs and the selected action.

**CB.UPDATE: Cost Estimation Techniques.** The update procedure for our contextual bandit oracles takes an example $\boldsymbol{x}$, action $a$, action probability $p$ and cost $c$ as input and updates its policy. We do this by reducing to a cost-sensitive classification oracle (Appendix C), which expects an example $\boldsymbol{x}$ and a cost vector $\boldsymbol{y} \in \mathbb{R}^K$ that specifies the cost for *all* actions (not just the selected one). The reduction challenge is constructing this cost-sensitive classification example given the input to CB.UPDATE. We consider three methods: inverse propensity scoring (Horvitz & Thompson, 1952), doubly robust estimation (Dudík et al., 2014) and multitask regression (Langford & Agarwal, 2017).

*Inverse Propensity Scoring (IPS):* IPS uses the selected action probability $p$ to correct for the shift in action proportions predicted by the policy $\pi^{\text{learn}}$. IPS estimates the target cost vector $\boldsymbol{y}$ as: $\boldsymbol{y}(i) = \frac{c}{p}\mathbf{1}[i = a]$, where $\mathbf{1}$ is an indicator function and where $a$ is the selected action and $c$ is the observed cost. While IPS yields an unbiased estimate of costs, it typically has a large variance as $p \to 0$.

*Doubly Robust Cost Estimation (DR):* The doubly robust estimator uses both the observed cost $c$ as well as its own predicted costs $\hat{y}(i)$ for *all* actions, forming a target that combines these two sources of information. DR estimates the target cost vector $\boldsymbol{y}$ as: $\boldsymbol{y}(i) = \hat{y}(i) + \mathbf{1}[i = a](c - \hat{y}(i))/p$. The DR estimator remains unbiased, and the estimated loss $\boldsymbol{y}$ helps decrease its variance.

*Multitask Regression (MTR):* The multitask regression estimator functions somewhat differently from IPS and DR. Instead of reducing to to cost-sensitive classification, MTR reduces directly to importance-weighted regression. MTR maintains $K$ different regressors for predicting costs given input/action pairs. Given $\boldsymbol{x}, a, c, p$, MTR constructs a *regression* example, whose input is $(\boldsymbol{x}, a)$, whose target output is $c$ and whose importance weight is $1/p$.

**CB.ACT: Exploration Strategies.** We experiment with three exploration strategies:

*Uniform:* explores randomly with probability $\epsilon$ and otherwise acts greedily (Sutton & Barto, 1998).

*Boltzmann:* varies action probabilities where action $a$ is chosen with probability proportional to $\exp[-c(a)/\text{temp}]$, where temp $\in \mathbb{R}^+$ is the temperature, and $c(a)$ is the predicted cost of action $a$.

*Bootstrap Exploration:* (Agarwal et al., 2014) trains a bag of multiple policies simultaneously. Each policy in the bag votes once on its predicted action, and an action is sampled from this distribution. To train, each example gets passed to each policy Poisson($\lambda = 1$)-many times, which ensures diversity . Bootstrap can operate in "greedy update" and "greedy prediction" mode (Bietti et al., 2017). In greedy update, we always update the first policy in the bag exactly once. In greedy prediction, we always predict the action from the first policy during exploitation.

## 3.3 THEORETICAL ANALYSIS

For simplicity, we first consider the case where we have access to a good reference policy $\pi^{\text{ref}}$ but do not have access to good Q-value estimates under $\pi^{\text{ref}}$. The only way one can obtain a Q-value estimate is to do a roll-out, but in a non-resettable environment, we can only do this once. We will subsequently consider the case of suboptimal (or missing) reference policies, in which the goal of the analysis will change from competing with $\pi^{\text{ref}}$ to competing with both $\pi^{\text{ref}}$ and a local optimality guarantee.

**Theorem 1.** *Setting $\beta = 1$, running* RESLOPE *for $N$ episodes with a contextual bandit algorithm, the average returned policy $\bar{\pi} = \mathbb{E}_n \pi_n$ has regret equal to the suboptimality of $\pi^{ref}$, namely:*

$$Regret(\bar{\pi}) \leq Regret(\pi^{ref}) + \frac{1}{N}\epsilon_{CB}(N) + \epsilon_{approx} \tag{1}$$

*where $\epsilon_{CB}(N)$ is the cumulative regret of the underlying contextual bandit algorithm after $N$ steps, and $\epsilon_{approx}$ is an approximation error term that goes to zero as $N \to \infty$ so long as the contextual bandit algorithm is no-regret and assuming all costs are realizable under the hypothesis class used by* RESLOPE.

In particular, when the problem is realizable and the contextual bandit algorithm is no-regret, RES-LOPE is also no-regret. The realizability assumption is unfortunate, but does not appear easy to remove (see Appendix D for the proof).

In the case that $\pi^{\text{ref}}$ is not known to be optimal, or not available, we follow the LOLS analysis and obtain a regret to a convex combination of $\pi^{\text{ref}}$ and the learned policy's one-step deviations (a form of local optimality) and can additionally show the following (proof in Appendix E):

**Theorem 2.** *For arbitrary $\beta$, define the combined regret of $\bar{\pi}$ as: $Regret_\beta(\bar{\pi}) = \beta[J(\bar{\pi}) - J(\pi^{ref})] + (1-\beta)\sum_h[J(\bar{\pi}) - \min_{\pi \in \Pi} \mathbb{E}_{s \sim d^h_{\bar{\pi}}} Q^{\bar{\pi}}(s, \pi)]$. The first term is suboptimality to $\pi^{ref}$; the second term is suboptimality to the policy's own realizable one-step deviations. Given a contextual bandit learning algorithm, and under a realizability assumption, the combined regret of $\bar{\pi}$ satisfies: $Regret_\beta(\bar{\pi}) \leq \frac{1}{N}\epsilon_{CB}(N) + \epsilon_{approx}$*

Again, if the contextual bandit algorithm is no regret, then $\epsilon_{\text{CB}}/N \to 0$ as $N \to \infty$; see Appendix E for the proof.

## 3.4 MULTI-DEVIATION RESIDUAL LOSS PREDICTION

Finally, we present the multiple deviation variant of RESLOPE. Algorithm 2 shows how RESLOPE operates under multiple deviations. The difference between the single and multiple deviation mode is twofold: 1. Instead of deviating at a single time step, *multi*-dev RESLOPE performs deviations at each time step in the horizon; 2. Instead of generating a single contextual bandit example per episode, *multi*-dev RESLOPE generates $H$ examples, where $H$ is the length of the time horizon, effectively updating the policy $H$ times.

These two changes means that we update the learned policy $\pi^{\text{learn}}$ multiple times per episode. Empirically, we found this to be crucial for achieving superior performance. Although, the generated samples for the same episode are not independent, this is made-up for by the huge increase in the

---

**Algorithm 2** RESIDUAL LOSS PREDICTION (RESLOPE) with *multiple* deviations

---

**Require:** Contextual bandit oracle CB, MAKEENVIRONMENT to build new enviornments
  1: Initialize a policy $\pi_0^{\text{learn}}$ {either randomly or from a pretrained model}
  2: **for all** episodes $t = 1 \ldots T$ **do**
  3:    env $\leftarrow$ MAKEENVIRONMENT($t$)
  4:    Initialize variables: examples $\boldsymbol{x}_h^{\text{dev}}$, actions $a_h^{\text{dev}}$, probabilities $p_h^{\text{dev}}$
        and costs $\hat{c}_h^{\text{dev}} = 0$ for $h = 1 \ldots \text{env}.H$
  5:    **for all** time steps $h = 1 \ldots \text{env}.H$ **do**
  6:        $\boldsymbol{x}_h^{\text{dev}} \leftarrow \text{env}.\text{STATEFEATURES}$ {computed by an RNN}
  7:        $(a_h^{\text{dev}}, p_h^{\text{dev}}) \leftarrow \text{CB}.\text{ACT}(\pi^{\text{learn}}, \boldsymbol{x}_h^{\text{dev}})$
  8:        $\hat{c}_h^{\text{dev}} \leftarrow \text{CB}.\text{COST}(\pi_{t-1}^{\text{learn}}, \boldsymbol{x}_h^{\text{dev}}, a_h^{\text{dev}})$
  9:        env.STEP($a_h^{\text{dev}}$) {updates environment and internal state of the RNN }
10:    **end for**
11:    $\ell_h^{\text{residual}} \leftarrow \text{env}.\text{FINALLOSS} - \sum_{h' \neq h} \hat{c}^{\text{dev}}(h')$ for all $h$
12:    $\pi_t^{\text{learn}} \leftarrow \text{CB}.\text{UPDATE}(\pi_{t-1}^{\text{learn}}, \boldsymbol{x}_h^{\text{dev}}, a_h^{\text{dev}}, p_h^{\text{dev}}, \ell_h^{\text{residual}})$ for all $h$
13: **end for**
14: Return average policy $\bar{\pi} = \frac{1}{T} \sum_t \pi_t^{\text{learn}}$

---

number of available samples for training (i.e. $T \times H$ samples for multiple deviations versus only $T$ samples in the single deviation mode). The theoretical analysis that precedes still holds in this case, but only makes sense when $\beta = 0$ because there is no longer any distinction between roll-in and roll-out, and so the guarantee reduces to a local optimality guarantee.

## 4 EXPERIMENTAL SETUP

We conduct experiments on both reinforcement learning and structured prediction tasks. Our goal is to evaluate how quickly different learning algorithms learn from episodic loss. We implement our models on top of the DyNet neural network optimization package (Neubig et al., 2017). [4]

**Reinforcement Learning Environments** We perform experiments in four standard reinforcement learning environments: Blackjack (classic card game), Hex (two-player board game), Cartpole (aka "inverted pendulum") and Gridworld. Our implementations of these environments are described in Appendix F and largely follows the AI Gym (Brockman et al., 2016) implementations. We report results in terms of *cumulative loss*, where loss is $-1 \times reward$ for consistency with the loss-based exposition above and the loss-based evaluation of bandit structured prediction (§2.2).

**Bandit Structured Prediction Environments** We also conduct experiments on structured prediction tasks. The evaluation framework we consider is the fully online setup described in (§2.2), measuring the degree to which various algorithms can effectively improve by observing only the episodic loss, and effectively balancing exploration and exploitation. We learn from one structured example at a time and we do a single pass over the available examples. We measure performance in terms of average cumulative loss on the online examples as well as on a held-out evaluation dataset. The loss on the online examples measures how much the algorithm is penalized for unnecessary exploration. We perform experiments on the three tasks described in detail in Appendix G: English Part of Speech Tagging, English Dependency Parsing and Chinese Part of Speech Tagging.

### 4.1 COMPARATIVE ALGORITHMS

We compare against three common reinforcement learning algorithms: Reinforce (Williams, 1992) with a baseline whose value is an exponentially weighted running average of rewards; Proximal Policy Optimization (PPO) (Schulman et al., 2017); and Advantage Actor-Critic (A2C) (Mnih et al., 2016). For the structured prediction experiments, since the bandit feedback is simulated based on labeled data, we can also estimate an "upper bound" on performance by running a supervised

---

[4]The code is available at `https://github.com/hal3/macarico`, `https://github.com/hal3/reslope`

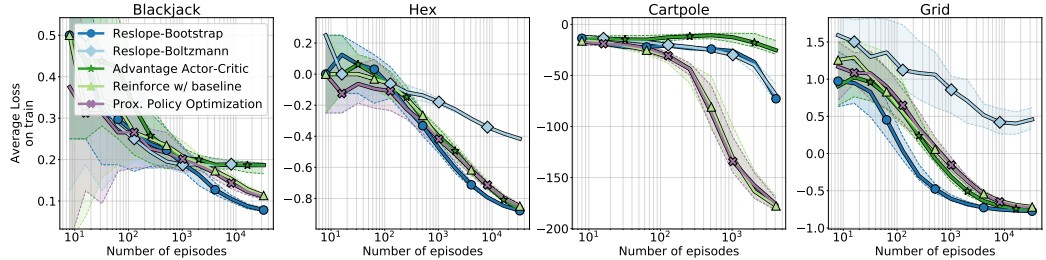

Figure 2: Average loss during learning on the four RL problems. Shaded regions are empirical quartiles over the experimental replicates with different random seeds.

learning algorithm that uses full information (thus forgoing issues of both exploration/exploitation and credit assignment). We run supervised DAgger to obtain such an upper bound.

## 4.2 POLICY ARCHITECTURE

In all cases, our policy is a recurrent neural network (Elman, 1990) that maintains a real-valued hidden state and combines: (a) its previous hidden state, (b) the features from the environment (described for each environment in the preceding sections), and (c) an embedding of its previous action. These form a new hidden state, from which a prediction is made. Formally, at time step $h$, $\boldsymbol{v}_h$ is the hidden state representation, $f(\text{state}_h)$ are the features from the environment and $a_h$ is the action taken. The recursion is:

$$\boldsymbol{v}_0 = \textbf{const} \qquad ; \qquad \boldsymbol{v}_{h+1} = ReLU\left(\mathbf{A}\left[\ \boldsymbol{v}_h\ ,\ \boldsymbol{f}(\text{state}_h)\ ,\ \textbf{emb}(a_h)\ \right]\right) \tag{2}$$

Here, $\mathbf{A}$ is a learned matrix, **const** is an initial (learned) state, **emb** is a (learned) action embedding function, and $ReLU$ is a rectified linear unit applied element-wise.

Given the hidden state $\boldsymbol{v}_h$, an action must be selected. This is done using a simple feedforward network operating on $\boldsymbol{v}_h$ with either no hidden layers (in which case the output vector is $\boldsymbol{o}_h = \mathbf{B}\boldsymbol{v}_h$) or a single hidden layer (where $\boldsymbol{o}_h = \mathbf{B}_2\, ReLU(\mathbf{B}_1\boldsymbol{v}_h)$). In the case of RESLOPE and DAgger, which expect *cost estimates* as the output of the policy, the output values $\boldsymbol{o}_h$ are used as the predicted costs (and $a_h$ might be the argmin of these costs when operating greedily). In the case of Reinforce, PPO and A2C, which expect action probabilities, these are computed as $softmax(-\boldsymbol{o}_h)$ from which, for instance, an action $a_h$ is sampled.

Details on optimization, hyperparameters and "deep learning tricks" are reported in Appendix H.

## 5 EXPERIMENTAL RESULTS

We study several questions empirically: 1. How does RESIDUAL LOSS PREDICTION compare to policy gradient methods on reinforcement learning and bandit structured prediction tasks? (§5.1) 2. What's the effect of ablating various parts of the RESLOPE approach, including multiple deviations? (§5.2) 3. Does RESLOPE succeed in learning a good representation of the loss? (§5.3)

### 5.1 REINFORCEMENT LEARNING AND BANDIT STRUCTURED PREDICTION RESULTS

In our first set of experiments, we compare RESLOPE to the competing approaches on the four reinforcement learning tasks described above. Figure 2 shows the results. In Blackjack, Hex and Grid, RESLOPE outperforms all the competing approaches with lower loss earlier in the learning process (though for Hex and Grid they all finish at the same near-optimal policy). For Cartpole, RESLOPE significantly *underperforms* both Reinforce and PPO.[5] Furthermore, in both Blackjack and Grid, the bootstrap exploration significantly improves upon Boltzmann exploration. In general, both RESLOPE performs quite well.[6]

---

[5]It is not entirely clear to us yet why this happens. We found that RESLOPE performs as well as Reinforce and PPO if we (a) replace the loss with one centered around zero and (b) replace the RNN policy with a simpler feed-forward network, but we do not include these results in the figure to keep the experiments consistent.

[6]In these experiments, PPO performs nearly identically to Reinforce. This happens because all of our experiments use a minibatch size of one. When PPO is run with a minibatch size of one, it reduces to *exactly*

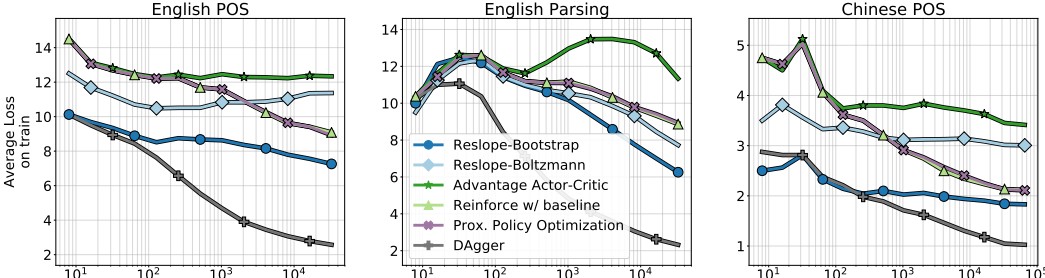

Figure 3: Average loss during learning for three bandit structured prediction problems. Also included are supervised learning results with DAgger.

In our second set of experiments, we compare the same algorithms *plus* the fully supervised DAgger algorithm on the three structured prediction problems; the results are in Figure 3. Here, we can observe RESLOPE significantly outperforming all alternative algorithms (except, of course, DAgger) on training loss (also on heldout (development) loss; see Figure 9 in the appendix). There is still quite a gap to fully supervised learning, but nonetheless RESLOPE is able to reduce training error significantly on all tasks: by over 25% on English POS, by about half on English dependency parsing, and by about 10% on Chinese POS tagging.

## 5.2 ABLATION OF RESIDUAL LOSS PREDICTION

In our construction of RESLOPE, there are several tunable parameters: which contextual bandit learner to use (IPS, DR, MTR), which exploration strategy (Uniform, Boltzmann, Bootstrap), and, for Bootstrap, whether to do greedy prediction and/or greedy update. In Table 1 (in the Appendix), we show the results on all tasks for ablating these various parameters. For the purpose of the ablation, we *fix* the "baseline" system as: DR, Bootstrap, and with both greedy prediction and greedy updates, though this is not uniformly the optimal setting (and therefore these numbers may differ slightly from the preceding figures). The primary take-aways from these results are: (1) MTR and DR are competitive, but IPS is *much* worse; (2) Bootstrap is much better than either other exploration method (especially uniform, not surprisingly); (3) Greedy prediction is a bit of a wash, with only small differences either way; (4) Greedy update is important. In Appendix I, we consider the effect of single vs multiple deviations and observe that significant importance of multiple deviations for all algorithms, with Reinforce and PPO behaving quite poorly with only single deviations.

## 5.3 EVALUATING THE LEARNED LOSS REPRESENTATION

In our final set of experiments, we study RESLOPE's performance under different—and especially non-additive—loss functions. Our goal is to investigate RESLOPE's ability to learn good representations for the episodic loss. We consider the following different incremental loss functions for each time step: Hamming (0/1 loss at each position), Time-Sensitive (cost for an error at position $h$ is equal to $h$) and Distance-Sensitive (cost for predicting $\hat{a}$ instead of $a$ is $|\hat{a} - a|$). To combine these per-stop losses into a per-trajectory loss $\boldsymbol{\tau}$ of length $H$, we compute the $H$-dimensional loss vector $\boldsymbol{\ell}$ suffered by RESLOPE along this trajectory. To consider both additive and non-additive combinations, we consider Lp norms of this loss vector. When the norm is L1, this is simple additive loss. More generally, we consider $\ell(\boldsymbol{\tau}) = \sqrt[p]{\sum_{t=1}^{t=H} \boldsymbol{\ell}^p(t)}$ for any $p > 0$.

Reinforce. We also have conducted experiments with PPO with larger minibatches; these results are reported in the appendix in Figure 8. In those experiments, we adjusted the minibatch size and number of epochs to match exactly with the PPO algorithm described in Schulman et al. (2017). In each iteration, each of $N$ actors collect $T$ timesteps of data. Then we construct the surrogate loss on these NT time steps of data, and optimize it with minibatch Adam for $K$ epochs. With these adjustments, PPO's performance falls between RESLOPE and Reinforce on Blackjack, slightly superior to RESLOPE on Hex, better than everything on Cartpole, and roughly equivalent to RESLOPE on Gridworld. We were, unfortunately, unable to conduct these experiments in the structured prediction setting, because the state memoization necessary to implement PPO with large/complex environments overflowed our system's memory quite quickly.

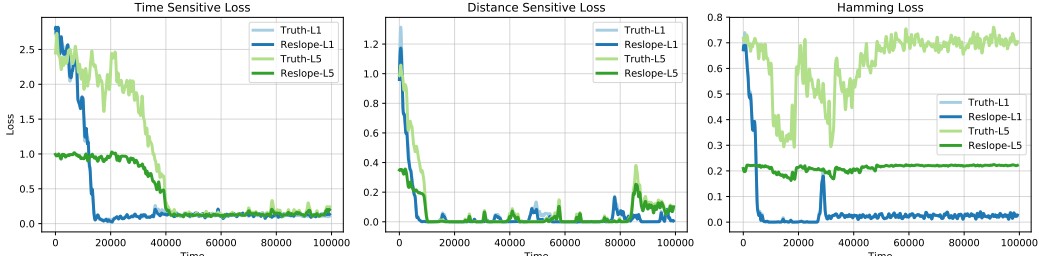

Figure 4: Empirical effect of additive vs non-additive loss functions. Performance is better when the loss is additive (blue) vs non-additive (green). The x-axis shows the number of episodes and the y-axis measures the incremental loss using the true loss function (light colors) and using RESLOPE (dark colors). If RESLOPE worked perfectly, these would coincide.

We run six different experiments using different incremental and episodic loss functions. For each incremental loss function (i.e. hamming, time sensitive, distance sensitive) we run two experiments: using the total hamming loss (additive) and an Lp norm of five (non-additive). Results are presented in Figure 4. We observe the following. RESLOPE can always learn the optimal representation for the incremental loss when the episodic loss function is additive. This is the case for all the three incremental loss functions: hamming, time sensitive, and distance sensitive. Learning is faster when the episodic loss function is additive. While RESLOPE is still able to learn a good representation even when using the L5 norm loss, this happens much later in comparison to the additive loss function (40k time steps for L5 norm vs 20k for total hamming loss). Not surprisingly, performance degrades as the episodic loss function becomes non-additive. This is most acute when using L-5 norm with the incremental hamming loss. This is expected as in the distance and time sensitive loss functions, RESLOPE observes a smoother loss function and learns to distinguish between different time steps based on the implicit encoding of time and distance information in the observed loss. RESLOPE can still learn a good representation for smoother episodic loss functions. This is shown empirically for time and distance sensitive loss functions.

## 6 RELATED WORK AND DISCUSSION

RESIDUAL LOSS PREDICTION builds most directly on the bandit learning to search frameworks LOLS (Chang et al., 2015) and BLS (Sharaf & Daumé, 2017). The "bandit" version of LOLS was analyzed theoretically but not empirically in the original paper; Sharaf & Daumé (2017) found that it failed to learn empirically.They addressed this by requiring additional feedback from the user, which worked well empirically but did not enjoy any theoretical guarantees. RESLOPE achieves the best of both worlds: a strong regret guarantee, good empirical performance, and no need for additional feedback. The key ingredient for making this work is using the residual loss structure together with strong base contextual bandit learning algorithms.

A number of recent algorithms have updated "classic" learning to search papers with deep learning underpinnings (Wiseman & Rush, 2016; Leblond et al., 2017). These aim to incorporate sequence-level global loss function to mitigate the mismatch between training and test time discrepancies, but only apply in the fully supervised setting. Mixing of supervised learning and reinforcement signals has become more popular in structured prediction recently, generally to do a better job of tuning for a task-specific loss using either Reinforce (Ranzato et al., 2015) or Actor-Critic (Bahdanau et al., 2016). The bandit variant of the structured prediction problem was studied by Sokolov et al. (2016), who proposed a reinforce method for optimizing different structured prediction models under bandit feedback in a log-linear structured prediction model.

A standard technique for dealing with sparse and episodic reward signals is reward shaping (Ng et al., 1999): supplying additional rewards to a learning agent to guide its learning process, beyond those supplied by the underlying environment. Typical reward shaping is hand-engineered; RESLOPE essentially learns a good task-specific reward shaping automatically. The most successful baseline approach we found is Proximal Policy Optimization (PPO, (Schulman et al., 2017)), a variant of Trust Region Policy Optimization (TRPO, (Schulman et al., 2015)) that is more practical.

Experimentally we have seen RESLOPE to typically learn more quickly than PPO. Theoretically both have useful guarantees of a rather incomparable nature.

Since RESLOPE operates as a reduction to a contextual bandit oracle, this allows it to continually improve as better contextual bandit algorithms become available, for instance work of Syrgkanis et al. (2016b) and Agarwal et al. (2014). Although RESLOPE is quite effective, there are a number of shortcomings that need to be addressed in future work. For example, the bootstrap sampling algorithm is prohibitive in terms of both memory and time efficiency. One approach for tackling this would be using the amortized bootstrap approach by Nalisnick & Smyth (2017), which uses amortized inference in conjunction with implicit models to approximate the bootstrap distribution over model parameters. There is also a question of whether the reduction to contextual bandits creates "reasonable" contextual bandit problems in conjunction with RNNs. While some contextual bandit algorithms assume strong convexity or linearity, the ones we employ operate on arbitrary policy classes, provided a good cost-sensitive learner exists. The degree to which this is true will vary by neural network architecture, and what can be guaranteed (e.g., no regret full-information online neural learning). A more significant problem in the multi-deviation setting is that as RESLOPE learns, the residual costs will change, leading to a shifting distribution of *costs*; in principle this could be addressed using CB algorithms that work in adversarial settings (Syrgkanis et al., 2016a;b), but largely remains an open challenge. RESLOPE is currently designed for discrete action spaces. Extension to continuous action spaces (Levine et al., 2016; Lillicrap et al., 2015) remains an open problem.

ACKNOWLEDGMENTS

We thank Paul Mineiro and the anonymous reviewers[7] for very helpful comments and insights (especially to reviewer #3 whose patient comments on the analysis section of this paper were incredibly helpful[8]). We also thank Khanh Nguyen, Shi Feng, Kianté Brantley, Moustafa Meshry, and Sudha Rao for reviewing earlier drafts for this work and Alekh Agarwal, Nan Jiang, and Adith Swaminathan for helpful discussions and comments. This work was partially funded by an Amazon Research Award. This material is based upon work supported by the National Science Foundation under Grant No. 1618193. Any opinions, findings, and conclusions or recommendations expressed in this material are those of the author(s) and do not necessarily reflect the views of the National Science Foundation.

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

## A MORE DETAILS ON CONTEXTUAL BANDIT ALGORITHMS

We assume that contexts are chosen i.i.d from an unknown distribution $\mathcal{D}(\boldsymbol{x})$, the actions are chosen from a finite action set $\mathcal{A}$, and the distribution over loss $\mathcal{D}(\ell|a, \boldsymbol{x})$ is fixed over time, but is unknown. In this context, the key challenge in contextual bandit learning is the exploration/exploitation problem. Classic algorithms for the contextual bandit problem such as EXP4.P (Beygelzimer et al., 2011) can achieve a $\sqrt{T}$ regret bound; in particular:

$$R\left(\text{EXP4}\right) \in O\left(\sqrt{TK\log|\Pi|}\right) \tag{3}$$

where $K = |A|$. When the regret is provably sublinear in $T$, such algorithms are often called "no regret" because their average regret per time step goes to zero as $T \to \infty$.

The particular contextual bandit algorithms we will use in this paper perform a second level of reduction: they assume access to an oracle supervised learning algorithm that can optimize a cost-sensitive loss, and transform the contextual bandit problem to a cost-sensitive classification problem. Algorithms in this family typically vary along two axes:

1. How to explore? I.e., faced with a new $\boldsymbol{x}$ how does the algorithm choose which action to take;

2. How to update? Given the observed loss $\ell_t$, how does the algorithm construct a supervised training example on which to train.

As a simple example, an algorithm might explore uniformly at random on $10\%$ of the examples and return the best guess action on $90\%$ of examples ($\epsilon$-greedy exploration). A single round to such an algorithm consists of a tuple $(\boldsymbol{x}, a, p)$, where $p$ is the probability with which the algorithm took action $a$. (In the current example, this would be $\frac{0.1}{K}$ for all actions except $\pi(\boldsymbol{x})$ and $0.9 + \frac{0.1}{K}$ for $a = \pi(\boldsymbol{x})$.) If the update rule were "inverse propensity scaling" (IPS) (Horvitz & Thompson, 1952), the generated cost-sensitive learning example would have $\boldsymbol{x}$ as an input, and a cost vector $\boldsymbol{c} \in \mathbb{R}^K$ with zeros everywhere except in position $a$ where it would take value $\frac{\ell}{p}$. The justification for this scaling is that in expectation over $a \sim p$, the expected value of this cost vector is equal to the true costs for each action. Neither of these choices is optimal (IPS has very high variance as $p$ gets small); we discuss alternative exploration strategies and variance reduction strategies (§3.2).

## B BANDIT STRUCTURED PREDICTION

Recently, it has become popular to solve structured prediction problems incrementally using some form of recurrent neural network (RNN) model. When the output $\boldsymbol{y}$ contains multiple parts (e.g., words in a translation), the RNN can predict each word in sequence, conditioning each prediction on all previous decisions. Although typically such models are trained to maximize cross-entropy with the gold standard output (in a fully supervised setting), there is mounting evidence that this has similar drawbacks to pre-RNN techniques, such as overfitting to gold standard prefixes (the model never learns what to do once it has made an error) and sensitivity to errors of different severity (due to error compounding).

By casting the structured prediction problem explicitly as a sequential decision making problem (Daumé & Marcu, 2005; Daumé et al., 2009; Ross et al., 2011; Neu & Szepesvári, 2009), we can avoid these problems by applying imitation-learning style algorithms to their solution. This "Learning to Search" framework (Figure 5) solves structured prediction problems by:

1. converting structured and control problems to search problems by defining a search space of states $\mathcal{S}$ and an action set $\mathcal{A}$;

2. defining structured features over each state to capture the inter-dependency between output variables;

3. constructing a reference policy $\pi^{\text{ref}}$ based on the supervised training data;

4. learning a policy $\pi^{\text{learn}}$ that imitates or improves upon the reference policy.

In the *bandit* structured prediction setting, this maps nicely to the type of MDPs described at the beginning of this section. The formal reduction, following (Daumé & Marcu, 2005) is to ignore the

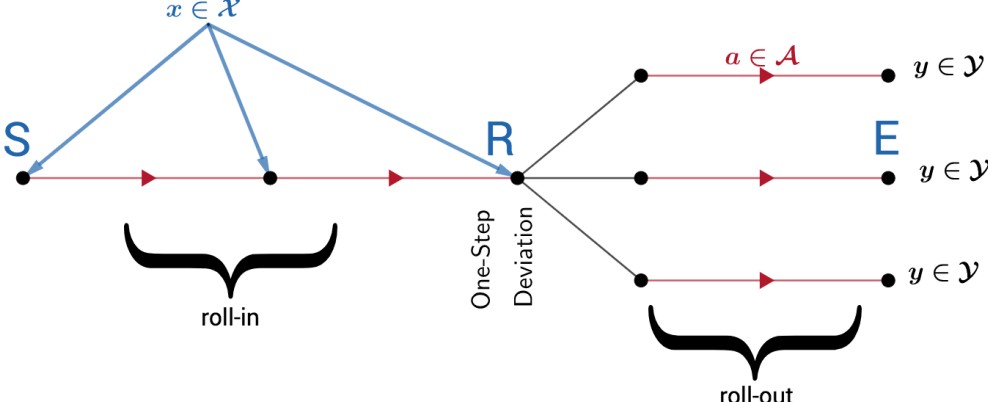

Figure 5: An example for a search space defined by a Learning to Search (L2S) algorithm. A search space is defined in terms of the set of states $\mathcal{X}$, and the set of actions $\mathcal{A}$. The agent starts at the initial state $S$, and queries the roll-in policy $\pi^{\text{in}}$ twice, next, at state $R$, the agent considers all three actions as possible one-step deviations. The agent queries the roll-out policy $\pi^{\text{out}}$ to generate three different trajectories from the set of possible output structures $\mathcal{Y}$.

first action $a_0$ and to transition to an "initial state" $s_1$ by drawing an input $\boldsymbol{x}^{\text{SP}} \sim \mathcal{D}^{\mathcal{X}}$. The search space of the structured prediction task then generates the remainder of the state/action space for this example. The episode terminates when a state, $s_H$ that corresponds to a "final output" is reached, at which point the structured prediction loss $\ell(\hat{\boldsymbol{y}}_{s_H} \mid \boldsymbol{x}^{\text{SP}})$ is computed on the output that corresponds to $s_H$. This then becomes the loss function $\mathcal{L}$ in the MDP. Clearly, learning a good policy under this MDP is equivalent to learning a structured prediction model with low expected loss.

## C   COST-SENSITIVE CLASSIFICATION

Many of the contextual bandit approaches we use in turn reduce the contextual bandit problem to a cost-sensitive classification problem. Cost-sensitive classification problems are defined by inputs $\boldsymbol{x}$ and cost vectors $\boldsymbol{y} \in \mathbb{R}^K$, where $\boldsymbol{y}(i)$ is the cost of choosing class $i$ on this example. The goal in cost-sensitive classification is to learn a classifier $f : \boldsymbol{x} \to [K]$ such that $\mathbb{E}_{(\boldsymbol{x}, \boldsymbol{y}) \sim \mathcal{D}}\big[\boldsymbol{y}(f(\boldsymbol{x}))\big]$ is small. A standard strategy for solving cost-sensitive classification is via reduction to regression in a one-against-all framework (Beygelzimer et al., 2005). Here, a regression function $g(\boldsymbol{x}, i) \in \mathbb{R}$ is learned that predicts costs given input/class pairs. A predicted class on an input $\boldsymbol{x}$ is chosen as $\operatorname{argmin}_i g(\boldsymbol{x}, i)$. This cost-sensitive one-against-all approach achieves low regret when the underlying regressor is good. In practice, we use regression against Huber loss.

## D   PROOF OF THEOREM 1

In a now-classic lemma, Kakade et al. (2003) and Bagnell et al. (2004) show that the difference in total loss between two policies can be computed exactly as a sum of per-time-step advantages of one over the other:

**Lemma 1** (Bagnell et al. (2004); Kakade et al. (2003)). *For all policies $\pi$ and $\pi'$:*

$$J(\pi) - J(\pi') = \sum_{h=1}^{H} \mathbb{E}_{s_h \sim d_\pi^h} \left[ Q^{\pi'}(s_h, \pi) - V^{\pi'}(s_h) \right] \tag{4}$$

*Proof of Theorem 1.* Let $\pi_n$ be the $n$th learned policy and $\bar{\pi}$ be the average learned policy. We wish to bound $J(\bar{\pi}) - J(\pi^*)$. We proceed as follows, largely following the AggreVaTe analysis (Ross &

Bagnell, 2014). We begin by noting that $J(\bar{\pi}) - J(\pi^*) = J(\bar{\pi}) - J(\pi^{\text{ref}}) + J(\pi^{\text{ref}}) - J(\pi^*)$ and will concern ourselves with bounding the first difference.

$$J(\bar{\pi}) - J(\pi^{\text{ref}}) = \mathbb{E}_n \sum_h \mathbb{E}_{s \sim d_{\pi_n}^h} \left[ Q^{\pi^{\text{ref}}}(s, \pi_n) - Q^{\pi^{\text{ref}}}(s, \pi^{\text{ref}}) \right] \tag{5}$$

Fix an $n$, and consider the sum above for a fixed deviation time step $h^{\text{dev}}$. In what follows, we consider $\pi_n$ to represent both the learned policy as well as the contextual bandit cost estimator, CB.COST.

$$\sum_h \mathbb{E}_{s \sim d_{\pi_n}^h} \left[ Q^{\pi^{\text{ref}}}(s, \pi_n) - Q^{\pi^{\text{ref}}}(s, \pi^{\text{ref}}) \right] \tag{6}$$

$$= \mathbb{E}_{s \sim d_{\pi_n}^{h^{\text{dev}}}} \left[ Q^{\pi^{\text{ref}}}(s, \pi_n) - Q^{\pi^{\text{ref}}}(s, \pi^{\text{ref}}) \right] + \sum_{h \neq h^{\text{dev}}} \mathbb{E}_{s \sim d_{\pi_n}^h} \left[ Q^{\pi^{\text{ref}}}(s, \pi_n) - Q^{\pi^{\text{ref}}}(s, \pi^{\text{ref}}) \right] \tag{7}$$

$$= \mathbb{E}_{s \sim d_{\pi_n}^{h^{\text{dev}}}} \left[ Q^{\pi^{\text{ref}}}(s, \pi_n) - \left( Q^{\pi^{\text{ref}}}(s, \pi^{\text{ref}}) - \sum_{h \neq h^{\text{dev}}} \mathbb{E}_{s' \sim d_{\pi_n}^h} \left[ Q^{\pi^{\text{ref}}}(s', \pi_n) - Q^{\pi^{\text{ref}}}(s', \pi^{\text{ref}}) \right] \right) \right] \tag{8}$$

$$= \mathbb{E}_{s \sim d_{\pi_n}^{h^{\text{dev}}}} \left[ Q^{\pi^{\text{ref}}}(s, \pi_n) - \left( \mathbb{E}_{s_H \sim \pi^{\text{ref}} \mid s_{h^{\text{dev}}} = s} \ell(s_H) - \sum_{h \neq h^{\text{dev}}} \mathbb{E}_{s' \sim d_{\pi_n}^h} \left[ \text{CB.COST}(\pi_n, s', \pi_n(s')) + \epsilon_{\text{approx}}(n, s') \right] \right) \right] \tag{9}$$

$$= \mathbb{E}_{s \sim d_{\pi_n}^{h^{\text{dev}}}} \left[ Q^{\pi^{\text{ref}}}(s, \pi_n) - \left( \mathbb{E}_{s_H \sim \pi^{\text{ref}} \mid s_{h^{\text{dev}}} = s} \ell(s_H) - \sum_{h \neq h^{\text{dev}}} \mathbb{E}_{s' \sim d_{\pi_n}^h} \text{CB.COST}(\pi_n, s', \pi_n(s')) \right) \right]$$
$$+ \sum_{h \neq h^{\text{dev}}} \mathbb{E}_{s' \sim d_{\pi_n}^h} \epsilon_{\text{approx}}(n, s') \tag{10}$$

$$= \mathbb{E}_{s \sim d_{\pi_n}^{h^{\text{dev}}}} \left[ Q^{\pi^{\text{ref}}}(s, \pi_n) - \text{Residual}(\pi_n, h^{\text{dev}}, s) \right] + \sum_{h \neq h^{\text{dev}}} \mathbb{E}_{s' \sim d_{\pi_n}^h} \epsilon_{\text{approx}}(n, s') \tag{11}$$

where $\text{Residual}(\pi_n, h^{\text{dev}}, s)$ is the estimated residual on this example.

Since the above analysis holds for an arbitrary $n$, it holds in expectation over $n$; thus:

$$J(\bar{\pi}) - J(\pi^{\text{ref}}) = \mathbb{E}_n \mathbb{E}_{s \sim d_{\pi_n}^{h^{\text{dev}}_n}} \left[ Q^{\pi^{\text{ref}}}(s, \pi_n) - \text{Residual}(\pi_n, h^{\text{dev}}, s) \right] + \mathbb{E}_n \sum_{h \neq h^{\text{dev}}_n} \mathbb{E}_{s' \sim d_{\pi_n}^h} \epsilon_{\text{approx}}(n, s') \tag{12}$$

$$= \frac{1}{N} \epsilon_{\text{CB}}(N) + \mathbb{E}_n \sum_{h \neq h^{\text{dev}}_n} \mathbb{E}_{s' \sim d_{\pi_n}^h} \epsilon_{\text{approx}}(n, s') \tag{13}$$

In the first line, the term in square brackets is exactly the cost being minimized by the contextual bandit algorithm and thus reduces to the regret of the CB algorithm.

In Eq (13), we have $H$-many regret minimizing online learners: one estimating the policy and one estimating estimating the $H - 1$-many costs. Cesa-Bianchi & Lugosi (2006) (Theorem 7.3) proves that in a $K$-player game, if each player minimizes its *internal regret*, then the overall values convergence in time-average to the value of the game. In order to apply this result to our setting we need to convert from external regret (which we are assuming about the underlying learners) to internal regret (which the theorem requires). This can be done using, for instance, the algorithm of which gives a general reduction from an algorithm that minimizes internal regret to one that minimizes external regret.

From there, by the strong realizability assumption, and the fact that multiple no-regret minimizers will achieve a time-averaged minimax value, we can conclude that as $N \to \infty$, the approximation error term will vanish. Moreover, the term in the round parentheses $(\dots)$ is exactly the expected value of the target of the contextual bandit cost. Therefore, If the CB algorithm has regret sublinear in $N$, both $\epsilon_{\text{CB}}(N)$ and the approximation error term go to zero as $N \to \infty$. This completes the proof that the overall algorithm is no-regret. $\qquad \square$

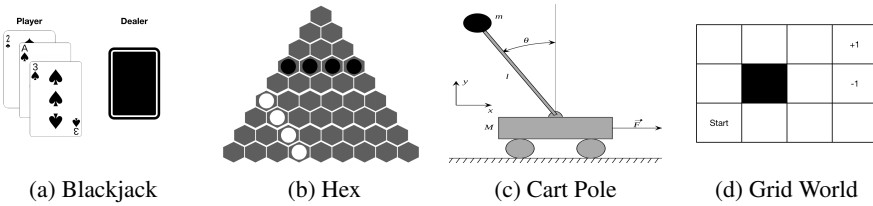

(a) Blackjack      (b) Hex      (c) Cart Pole      (d) Grid World

Figure 6: Reinforcement Learning Tasks

## E    PROOF OF THEOREM 2

*Proof of Theorem 2.* The proof follows a combination of the proof of Theorem 1 with the LOLS analysis. Using the same notation as before, additionally let $\pi_n^{\text{out}}$ be the mixture of $\pi_n$ with $\pi^{\text{ref}}$ for rollout.

First, we observe (LOLS Eq 6):

$$J(\bar{\pi}) - J(\pi^{\text{ref}}) = \mathbb{E}_n \sum_h \mathbb{E}_{s \sim d_{\pi_n}^h}[Q^{\pi^{\text{ref}}}(s, \pi_n) - Q^{\pi^{\text{ref}}}(s, \pi^{\text{ref}})] \tag{14}$$

Then (LOLS Eq 7):

$$\sum_h \left[ J(\bar{\pi}) - \min_{\pi \in \Pi} \mathbb{E}_{s \sim d_{\bar{\pi}}^h} Q^{\bar{\pi}}(s, \pi) \right] \leq \mathbb{E}_n \sum_h \mathbb{E}_{s \sim d_{\pi_n}^h} \left[ Q^{\pi_n}(s, \pi_n) - \min_a Q^{\pi_n}(s, a) \right] \tag{15}$$

So far nothing has changed. It will be convenient to define $Q_\beta^{\pi_n}(s) = \beta \min_a Q^{\pi^{\text{ref}}}(s, a) + (1 - \beta) \min_a Q^{\pi_n}(s, a)$. For each $n$ fix the deviation time step $h_n^{\text{dev}}$. We plug these together ala LOLS and get:

$$\beta \Big( J(\bar{\pi}) - J(\pi^{\text{ref}}) \Big) + (1 - \beta) \Big( J(\bar{\pi}) - \min_{\pi \in \Pi} \mathbb{E}_{s \sim d_{\bar{\pi}}^h} Q^{\bar{\pi}}(s, \pi) \Big) \tag{16}$$

$$\leq \mathbb{E}_n \sum_h \mathbb{E}_{s \sim d_{\pi_n}^h} \left[ Q^{\pi_n^{\text{out}}}(s, \pi_n) - \beta \min_a Q^{\pi^{\text{ref}}}(s, a) - (1 - \beta) \min_a Q^{\pi_n}(s, a) \right] \tag{17}$$

$$= \mathbb{E}_n \sum_h \mathbb{E}_{s \sim d_{\pi_n}^h} \left[ Q^{\pi_n^{\text{out}}}(s, \pi_n) - Q_\beta^{\pi_n}(s) \right] \tag{18}$$

$$= \mathbb{E}_n \mathbb{E}_{s^{\text{dev}} \sim d_{\pi_n}^{h_n^{\text{dev}}}} \left[ Q^{\pi_n^{\text{out}}}(s^{\text{dev}}, \pi_n) - \left( Q_\beta^{\pi_n}(s^{\text{dev}}) - \sum_{h \neq h_n^{\text{dev}}} \mathbb{E}_{s_h \sim d_{\pi_n}^h} \Big( Q^{\pi_n^{\text{out}}}(s_h, \pi_n) - Q_\beta^{\pi_n}(s_h) \Big) \right) \right] \tag{19}$$

$$= \mathbb{E}_n \mathbb{E}_{s^{\text{dev}} \sim d_{\pi_n}^{h_n^{\text{dev}}}} \left[ Q^{\pi_n^{\text{out}}}(s^{\text{dev}}, \pi_n) - \left( \mathbb{E}_{s_H \sim d_{\pi_n}^H \mid s_{h_n^{\text{dev}}} = s^{\text{dev}}} \mathcal{L}_n(s_H) - \sum_{h \neq h^{\text{dev}}} \text{CB.COST}(\pi_n, s_h) \right) \right] \tag{20}$$

The final step follows because the inner-most expectation is exactly what the contextual bandit algorithm is estimating, and $Q_\beta^{\pi_n}(s^{\text{dev}})$ is exactly the expectation of the observed loss. At this point the rest of the proof follows that of Theorem 1, relying on the same internal-to-external regret transformation, and the joint no-regret minimization of all "players." $\qquad\square$

## F    DETAILS ON REINFORCEMENT LEARNING ENVIRONMENTS

**Blackjack** is a card game where the goal is to obtain cards that sum to as near as possible to 21 without going over. Players play against a fixed dealer who hits until they have at least 17. Face cards (Jack, Queen, King) have a point value of 10. Aces can either count as 11 or 1, and a card is called "usable" at 11. The reward for winning is $+1$, drawing is 0, and losing is $-1$. The world is partially visible: the player can see one their own cards and one of the two initial dealer cards.

| Chinese POS | NT | | NN | | NN | | NN | | NN | | AD | |
|---|---|---|---|---|---|---|---|---|---|---|---|---|
| | 今年(this year) | 全球(global) | 手机(mobile) | 市场(market) | 规模(size) | 将(will) ... |

| English POS | NNP | NNP | , | CD | NNS | JJ | , | MD | VB | DT | NN | IN | DT | JJ | NN |
|---|---|---|---|---|---|---|---|---|---|---|---|---|---|---|---|
| | Pierre | Vinken | , | 61 | years | old | , | will | join | the | board | as | a | nonexecutive | director ... |

| Parsing | Root Flying planes can be dangerous |
|---|---|

Figure 7: Example inputs for part of speech tagging and dependency parsing.

**Hex** is a classic two-player board game invented by Piet Hein and independently by John Nash (Hayward & Van Rijswijck, 2006; Nash, 1952). The board is an $n \times n$ rhombus of hexagonal cells. Players alternately place a stone of their color on any empty cell. To win, a player connects her two opposing sides with her stones. We use $n = 5$; the world is fully visible to the agent, with each hexagon showing as unoccupied, occupied with white or occupied with black. The reward is $+1$ for winning and $-1$ for losing.

**Cart Pole** is a classic control problem variously referred to as the "cart-pole", "inverted pendulum", or "pole balancing" problem (Barto et al., 1983). Is is an example of an inherently unstable dynamic system, in which the objective is to control translational forces that position a cart at the center of a finite width track while simultaneously balancing a pole hinged on the cart's top. In this task, a pole is attached by a joint to a cart which moves along a frictionless track (Figure 6c). The system is controlled by applying a force of $+1$ or $-1$ to the cart, thus, we operate in a discrete action space with only two actions. The pendulum starts upright, and the goal is to prevent it from falling over. The episode ends when the pole is more than 15 degrees from the vertical axis, or the cart moves more than 2.4 units from the center. The state is represented by four values indicating the poles position, angle to the vertical axis, and the linear and angular velocities. The total cumulative reward at the end of the episode is the total number of time steps the pole remained upright before the episode terminates.

**Grid World** consists of a simple $3 \times 4$ grid, with a $+1$ reward in the upper-right corner and $-1$ reward immediately below it; the cell at $(1, 1)$ is blocked (Figure 6d). The agent starts at a random unoccupied square. Each step costs $0.05$ and the agent has a $10\%$ chance of misstepping. The agent only gets partial visibility of the world: it gets an indicator feature specifying which directions it can step. The only reward observed is the complete sum of rewards over an episode.

## G STRUCTURED PREDICTION DATA SETS

**English POS Tagging** we conduct POS tagging experiments over the 45 Penn Treebank (Marcus et al., 1993) tags. We simulate a domain adaptation setting by training a reference policy on the TweetNLP dataset (Owoputi et al., 2013) which achieves good accuracy in domain, but performs badly out of domain. We simulate bandit episodic loss over the entire Penn Treebank Wall Street Journal (sections $02 \rightarrow 21$ and 23), comprising 42k sentences and about one million words. The measure of performance is the average Hamming loss. We define the search space by sequentially selecting greedy part-of-speech tags for words in the sentence from left to right.

**Chinese POS Tagging** we conduct POS tagging experiments over the Chinese Penn Treebank (3.0) (Xia, 2000) tags. We simulate a domain adaptation setting by training a reference policy on the Newswire domain from the Chinese Treebank Dataset (Xue et al., 2005) and simulate bandit episodic feedback from the spoken conversation domain. We simulate bandit episodic loss over 40k sentences and about 300k words. The measure of performance is the average Hamming loss. We define the search space by sequentially selecting greedy part-of-speech tags for words in the sentence from left to right.

**English Dependency Parsing** For this task, we assign a grammatical head (i.e. parent) for each word in the sentence. We train an arc-eager dependency parser (Nivre, 2003) which chooses among

(at most) four actions at each state: Shift, Reduce, Left or Right. The reference policy is trained on the TweetNLP dataset and evaluated on the Penn Treebank corpus. The loss is the unlabeled attachment score (UAS), which measures the fraction of words that are assigned the correct parent.

In all structured prediction settings, the feature representation begins with pretrained (and non-updated) embeddings. For English, these are the 6gb Glove embeddings (Pennington et al., 2014); for Chinese, these are the FastText embeddings (Joulin et al., 2016). We then run a bidirectional LSTM (Hochreiter & Schmidhuber, 1997) over the input sentence. The input features for labeling the $n$th word in POS tagging experiments are the biLSTM representations at position $n$. The input features for dependency actions are a concatenation of the biLSTM features of the next word on the buffer and the two words on the top of the stack.

## H    OPTIMIZATION, HYPERPARAMETER SELECTION AND "TRICKS"

We optimize all parameters of the model using the Adam[9] optimizer (Kingma & Ba, 2014), with a tuned learning rate, a moving average rate for the mean of $\beta_1 = 0.9$ and for the variance of $\beta_2 = 0.999$; epsilon (for numerical stability) is fixed at $1e - 8$ (these are the DyNet defaults). The learning rate is tuned in the range $\{0.050.01, 0.005, 0.001, 0.0005, 0.0001\}$.

For the structured prediction experiments, the following input features hyperparameters are tuned:

- Word embedding dimension $\in \{50, 100, 200, 300\}$ (for the Chinese embeddings, which come only in 300 dimensional versions, we took the top singular vectors to reduce the dimensionality).
- BiLSTM dimension $\in \{50, 150, 300\}$
- Number of BiLSTM layers $\in \{1, 2\}$
- Pretraining: DAgger or AggreVaTe initialization with probability of rolling in with the reference policy $\in \{0.0, 0.999^N, 0.99999^N, 1.0\}$, where $N$ is the number of examples
- Policy RNN dimension $\in \{50, 150, 300\}$
- Number of policy layers $\in \{1, 2\}$
- Roll-out probability $\beta \in \{0.0, 0.5, 1.0\}$

For each task, the network architecture that was optimal for supervised pretraining was fixed and used for all bandit learning experiments[10].

For the reinforcement learning experiments, we tuned:

- Policy RNN dimension $\in \{20, 50, 100\}$
- Number of policy layers $\in \{1, 2\}$

Some parameters we do not tune: the nonlinearities used, the size of the action embeddings (we use 10 in all cases), the input RNN form for the text experiments (we always use LSTM instead of RNN or GRU based on preliminary experiments). We do not regularize our models (weight shrinkage only reduced performance in initial experiments) nor do we use dropout. Pretraining of the structured prediction models ran for 20 passes over the data with early stopping based on held-out loss. The state of the optimizer was reset once bandit learning began.

The variance across difference configurations was relatively small across RL tasks, so we chose a two layer policy with 20 dimensional vectors for all RL tasks.

Each algorithm also has a set of hyperparameters; we tune them as below:

- Reinforce: with baseline or without baseline

---

[9]We initially experimented also with RMSProp (Tieleman & Hinton, 2012) and AdaGrad (Duchi et al., 2011) but Adam consistently performed as well or better than the others on all tasks.

[10]English POS tagging and dependency parsing: DAgger $0.99999^N$, 300 dim embeddings, 300 dim 1 layer LSTM, 2 layer 300 dimensional policy; Chinese POS tagging: DAgger $0.999^N$, 300 dim embeddings, 50 dim 2 layer LSTM, 1 layer 50 dimensional policy).

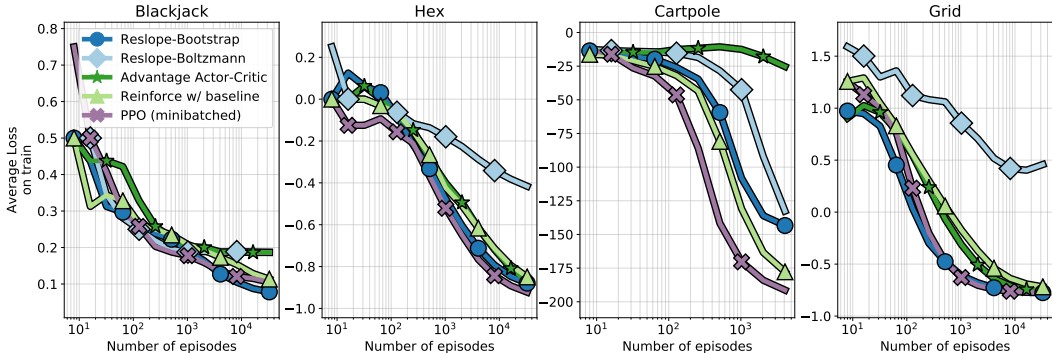

Figure 8: Average loss during learning on the four RL problems, including PPO with minibatching. (None of the other algorithms use minibatching, so the comparison is somewhat unfair.)

|  | **Reinforcement Learning** | | | | **Bandit SP** | | |
|---|---|---|---|---|---|---|---|
|  | **Blackjack** | **Cartpole** | **Grid** | **Hex** | **Zh-POS** | **En-Dep** | **En-POS** |
| total loss | 0.17 | $-28.0$ | 0.69 | $-0.88$ | 1.8 | 6.3 | 7.3 |
| loss std | 0.021 | 23.0 | 0.74 | 0.008 | 0.019 | 0.58 | 0.77 |
| $\rightarrow$ MTR | $-1.55$ | $-0.105$ | $-0.783$ | 2.88 | 0.023 | 1.56 | 0.661 |
| $\rightarrow$ IPS | $-1.81$ | 0.77 | $-0.28$ | 0.427 | 282.0 | 13.2 | 17.6 |
| $\rightarrow$ Boltzmann | 2.85 | 0.263 | 0.184 | 54.8 | 275.0 | 14.1 | 18.3 |
| $\rightarrow$ Uniform | 10.8 | 0.28 | 0.566 | 104.0 | 285.0 | 16.1 | 13.8 |
| $-$ g-predict | $-0.638$ | 0.362 | $-0.31$ | $-0.151$ | 0.236 | 0.314 | 0.596 |
| $-$ g-update | 1.03 | 0.508 | $-0.158$ | 2.24 | 7.11 | 3.87 | 2.79 |

Table 1: Results of ablating various parts of the RESIDUAL LOSS PREDICTION approach. Columns are tasks. The first two rows are the cumulative average loss over multiple runs and its standard deviation. The numbers in the rest of the column measure how much it hurts (positive number) or helps (negative number) to ablate the corresponding parameter. To keep the numbers on a similar scale, the changes are reported as *multiples* of the standard deviation. So a value of 2.0 means that the cumulative loss gets worse by an additive factor of two standard deviations.

- A2C: a multiplier on the relative importance of actor loss and critic loss $\in$ $\{0.1, 0.2, 0.5, 1.0, 2.0, 5.0, 10.0\}$

- PPO: with baseline or without baseline; and epsilon parameter $\in$ $\{0.01, 0.05, 0.1, 0.2, 0.4, 0.8\}$

- RESLOPE: update strategy (IPS, DR, MTR) and exploration strategy (uniform, Boltzmann or Bootstrap)

In each reinforcement/bandit experiment, we *optimistically* pick algorithm hyperparameters and learning rate based on final evaluation criteria, noting that this likely provides unrealistically optimistic performance for *all* algorithms. We perform 100 replicates of every experiment in the RL setting and 20 replicates in the structured prediction setting. We additionally ablate various aspects of RESLOPE in §5.2.

We employ only two "tricks," both of which are defaults in dynet: gradient clipping (using the default dynet settings) and smart parameter initialization (dynet uses Glorot initialization (Glorot & Bengio, 2010)).

## I  EFFECT OF SINGLE VS MULTIPLE DEVIATIONS

Next, we consider the single-deviation version of RESLOPE (1) versus the multiple-deviation version (2). To enable comparison with alternative algorithms, we also experiment with variants of Reinforce, PPO and DAgger that are only allowed single deviations as well (also chosen uniformly

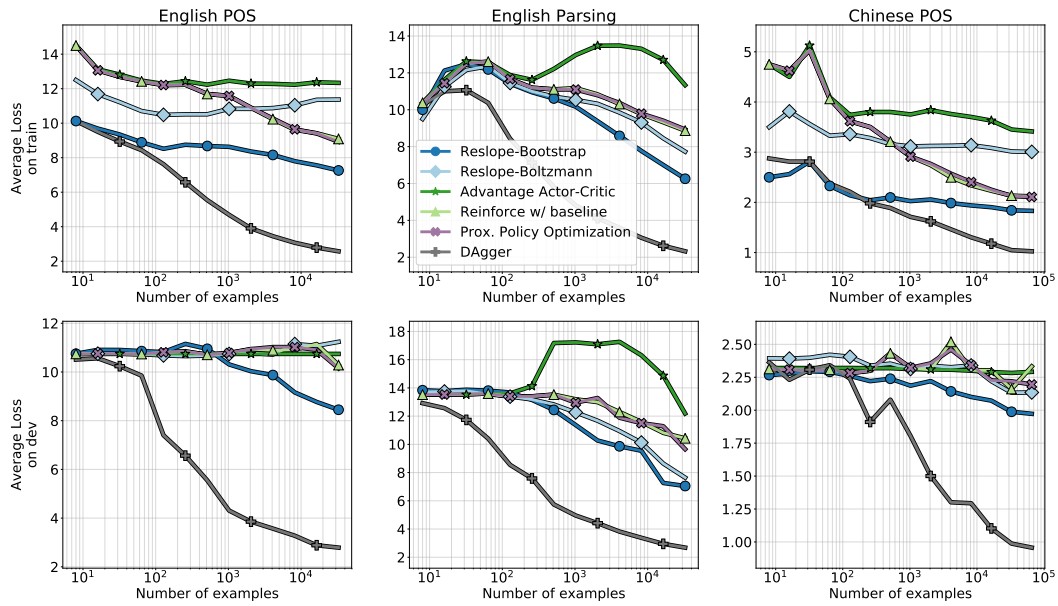

Figure 9: Average loss (top) and heldout loss (bottom) during learning for three bandit structured

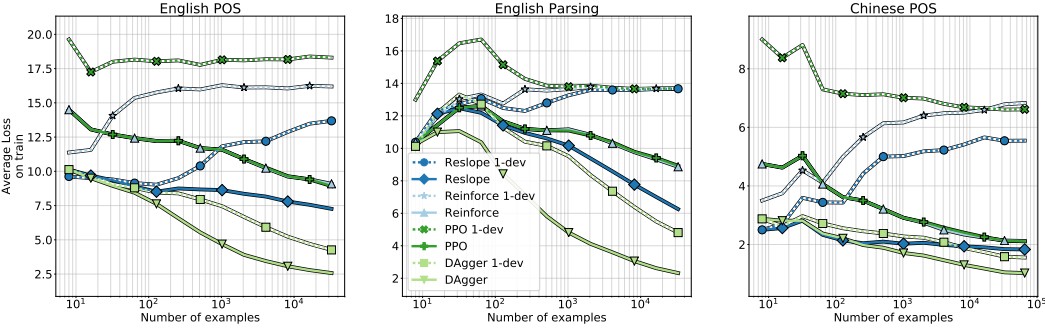

Figure 10: The empirical effect of multiple deviations for different algorithms.

at random). The results are shown in Figure 10. Not surprisingly, all algorithms suffer when only allowed single deviations. PPO makes things worse over time (likely because its updates are very conservative, such that even in the original PPO paper the authors advocate multiple runs over the same data), as does Reinforce. DAgger still learns, though more slowly, when only allowed a single deviation. RESLOPE behaves similarly though not quite as poorly. Overall, this suggests that even though the samples generated with multiple deviations by RESLOPE are no longer independent, the gain in number of samples more than makes up for this.

## J  SYNTHETIC DATA FOR EVALUATING THE LEARNED LOSS REPRESENTATION

Experiments were conducted on a synthetic sequence labeling dataset. Input sequences are random integers (between one and ten) of length 6. The ground truth label for the $h$th word is the corresponding input mod 4. We generate 16k training sequences for this experiment. We run RESLOPE with bootstrap sampling in multiple deviation mode. We use the MTR cost estimator, and optimize the policies using ADAM with a learning rate of $0.01$.

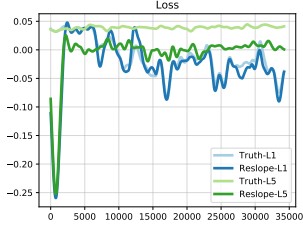

Figure 11: Empirical effect of additive vs non-additive loss functions. Performance is better when the loss is additive (blue) vs non-additive (green). The x-axis shows the number of episodes and the y-axis measures the incremental loss using the true loss function (light colors) and using RESLOPE (dark colors). If RESLOPE worked perfectly, these would coincide.

## K   EVALUATING THE LEARNED LOSS REPRESENTATION FOR GRID WORLD

In this section, we study RESLOPE's performance under different—and especially non-additive—loss functions. This experiment is akin to the experimental setting in section 5.3, however it's performed on the grid world reinforcement learning environment, where the quantitative aspects of the loss function is well understood.

We study a simple 4×4 grid, with a $+1$ reward in the upper-right corner and $-1$ reward immediately below it; the cells at $(1, 1)$ and $(2, 1)$ are blocked. The agent starts at a random position in the grid. Each step costs $+0.05$ and the probability of success is $0.9$. The agent has full visibility of the world: it knows its horizontal and vertical position in the grid.

We consider two different episodic reward settings:

1. The only reward observed is the complete sum of losses over an episode. (additive setting);

2. The only reward observed is the L5 norm of the vector of losses over an episode (non-additive setting).

Results are shown in Figure 11. Results are very similar to the structured prediction setting (section 5.3). Performance is better when the loss is additive (blue) vs non-additive (green).

