# OpenReview forum: "Residual Loss Prediction: Reinforcement Learning With No Incremental Feedback"
_ICLR.cc/2018/Conference — Accept (Poster)_

### Official Review · AnonReviewer2 · 2017-11-23
**RESIDUAL LOSS PREDICTION: episodic RL and learning to search without incremental feedback**

**Rating:** 7
**Confidence:** 2

**Review:**

The authors propose a new episodic reinforcement learning algorithm based on contextual bandit oracles.
The key specificity of this algorithm is its ability to deal with the credit assignment problem by learning automatically a progressive "reward shaping" (the residual losses) from a feedback that is only provided at the end of the epochs.

The paper is dense but well written.

The theoretical grounding is a bit thin or hard to follow.
The authors provide a few regret theoretical results (that I did not check deeply) obtained by reduction to "value-aware" contextual bandits.

The experimental section is solid. The method is evaluated on several RL environments against state of the art RL algorithms. It is also evaluated on bandit structured prediction tasks.
An interesting synthetic experiment (Figure 4) is also proposed to study the ability of the algorithm to work on both decomposable and non-decomposable structured prediction tasks.


Question 1: The credit assignment approach you propose seems way more sophisticated than eligibility traces in TD learning. But sometimes old and simple methods are not that bad. Could you develop a bit on the relation between RESLOPE and eligibility traces ?

Question 2: RESLOPE is built upon contextual bandits which require a stationary environment. Does RESLOPE inherit from this assumption?


Typos:
page 1
"scalar loss that output." -> "scalar loss."
", effectively a representation" -> ". By effective we mean effective in term of credit assignment."
page 5
"and MTR" -> "and DR"
page 6
"in simultaneously." -> ???
".In greedy" -> ". In greedy"

---

> ### Author Response · Authors · 2018-01-05
> **Authors' Response To AnonReviewer2**
>
> Question 1: RESLOPE and Eligibility Traces
>
> Both RESLOPE and eligibility trace algorithms tackles the problem of credit assignment when learning by interaction with the environment. In eligibility trace algorithms, e.g. TD(λ), a state is eligible for credit assignment if it was recently visited, with the eligibility declining over time [1]. In our episodic setting, our notion of eligibility decay is "the end of the episode": any reward from this episode is eligible, and reward from other episodes is not. The "degree" of eligibility is most similar to the probability of the exploration event which created the observation (the deviation). This is particularly important for getting unbiased & convergent estimates.
>
> Question 2: RESLOPE and Non-stationary Environments
>
> Thank you for raising this point: we were remiss to not include this in the initial draft and have now added a bit of discussion in the last section. The issue pointed out here is that because the policy is changing, the reward decomposition is changing, so the costs that the CB algorithm sees are also changing. While many CB algorithms operate effectively under shifting distributions of x (e.g. most online CB algorithms), many cannot work with the "label distribution" shifts. There has been some work on CB in an adversarial environment, but to our knowledge none of these algorithms is efficient. It seems likely that the RESLOPE setting is probably not as bad as full adversarial, and perhaps something could be done in the middle, but this is still an open question.
>
> [1] Satinder P. Singh and Richard S. Sutton, Reinforcement learning with replacing eligibility traces, pp. 123–158, Springer US, Boston, MA, 1996.

---

### Official Review · AnonReviewer3 · 2017-11-27
**Solid paper, many confusions in the original paper have been answered**

**Rating:** 7
**Confidence:** 5

**Review:**

After reading the other reviews and the authors' responses, I am satisfied that this paper is above the accept threshold.  I think there are many areas of further discussion that the authors can flesh out (as mentioned below and in other reviews), but overall the contribution seems solid.   I also appreciate the reviewers' efforts to run more experiments and flesh out the discussion in the revised version of the submission.

Final concluding thoughts:
-- Perhaps pi^ref was somehow better for the structured prediction problems than RL problems?
-- Can one show regret bound for multi-deviation if one doesn't have to learn x (i.e., we were given a good x a priori)?



---------------------------------------------
ORIGINAL REVIEW

First off, I think this paper is potentially above the accept threshold.  The ideas presented are interesting and the results are potentially interesting as well.   However, I have some reservations, a significant portion of which stem from not understanding aspects of the proposed approach and theoretical results, as outlined below.



The algorithm design and theoretical results in the appendix could be made substantially more rigorous. Specifically:

--  basic notations such as regret (in Theorem 1), the total reward (J), Q-value (Q), and value function (V) are not defined.  While these concepts are fairly standard, it would be highly beneficial to define them formally.

-- I'm not convinced that the "terms in the parentheses" (Eq. 7) are "exactly the contextual bandit cost".  I would like to see a more rigorous derivation of the connection.  For instance, one could imagine that the policy disadvantage should be the difference between the residual costs of the bandit algorithm and the reference policy, rather than just the residual cost of the bandit algorithm.

-- I'm a little unclear in the proof of Theorem 1 where Q(s,pi_n) from Eq 7 fits into Eq 8.

-- The residual cost used for CB.update depends on estimated costs at other time steps h!=h_dev.  Presumably, these estimated costs will change as learning progresses.  How does one reconcile that?  I imagine that it could somehow work out using a bandit algorithm with adversarial guarantees, but I can also imagine it not working out.  I would like to see a rigorous treatment of this issue.

-- It would be nice to see an end-to-end result that instantiates Theorem 1 (and/or Theorem 2) with a contextual bandit algorithm to see a fully instantiated guarantee.



With regards to the algorithm design itself, I have some confusions:

-- How does one create x in practice? I believe this is described in Appendix H, but it's not obvious.

-- What happens if we don't have a good way to generate x and it must be learned as well?  I'd imagine one would need larger RNNs in that case.

-- If x is actually learned on-the-fly, how does that impact the theoretical results?

-- I find it curious that there's no notion of future reward learning in the learning algorithm.  For instance, in Q learning, one directly models the the long-term (discounted) rewards during learning.  In fact, the theoretical analysis talks about advantage functions as well.  It would be nice to comment on this aspect at an intuitive level.



With regards to the experiments:

-- I find it very curious that the results are so negative for using only 1-dev compared to multi-dev (Figure 9 in Appendix).  Given that much of the paper is devoted to 1-dev, it's a bit disappointing that this issue is not analyzed in more detail, and furthermore the results are mostly hidden in the appendix.

-- It's not clear if a reference policy was used in the experiments and what value of beta was used.

-- Can the authors speculate about the difference in performance between the RL and bandit structured prediction settings?  My personal conjecture is that the bandit structured prediction settings are more easily decomposable additively, which leads to a greater advantage of the proposed approach, but I would like to hear the authors' thoughts.



Finally, the overall presentation of this paper could be substantially improved.  In addition to the above uncertainties, some more points are described below.  I don't view these points as "deal breakers" for determining accept/reject.

-- This paper uses too many examples, from part-of-speech tagging to credit assignment in determining paths.  I recommend sticking to one running example, which substantially reduces context switching for the reader.  In every such example, there are extraneous details are not relevant to making the point, and the reader needs to spend considerable effort figuring that out for each example used.

-- Inconsistent language. For instance, x is sometimes referred to as the "input example", "context" and "features".

-- At the end of page 4, "Internally, ReslopePolicy takes a standard learning to search step."  Two issues: 1) ReslopePolicy is not defined or referred to anywhere else.  2) is the remainder of that paragraph a description of a "standard learning to search step"?

-- As mentioned before, Regret is not defined in Theorem 1 & 2.

-- The discussion of the high-level algorithmic concepts is a bit diffuse or lacking.  For instance, one key idea in the algorithmic development is that it's sufficient to make a uniformly random deviation.  Is this idea from the learning to search literature?  If so, it would be nice to highlight this in Section 2.2.

---

> ### Author Response · Authors · 2018-01-05
> **Authors' Response to AnonReviewer3**
>
> How does RESLOPE create x?
>
> RESLOPE learns a representation for the input x on the fly using a neural  network architecture as described in Appendix H. We start off with a simple feature representation in all the problems and the model learns a better representation using a neural network architecture. We’d appreciate any comments regarding the clarity of this section and we’ll incorporate any suggestions in the final version.
>
> As a recap: For English POS tagging and dependency parsing we use 300 dimensional word embeddings, 300 dimensional 1 layer LSTM, and 2 layer 300 dimensional RNN policy; for the Chinese POS tagging: we use 300 dimensional word embeddings, 50 dimensional two layer LSTM, one layer 50 dimensional RNN policy. For reinforcement learning, we chose a two layer RNN policy with 20 dimensional vectors. We start off with a simple initial state representation and learn a better representation using the policy network. The initial state representation is task dependant. For instance, in cartpole, the state is represented by a four dimensional vector: [position of cart, velocity of cart, angle of pole, rotation rate of pole].
>
> What happens if we don't have a good way to generate x and it must be learned as well?
>
> This is the case in all our experiments. We start-off with simple features and learn a better representation on the fly using a neural network architecture. For structured prediction tasks, the simple features are just the word indices in the dictionary, we learn word embedding for these words and keep track of the state using an RNN architecture (as described above). For RL tasks we start off by simple features of the current state and feed these features to an RNN network to compute the final input x.
>
> If x is learned on the fly, how does that impact the theoretical results?
>
> In the single deviation case, one can think of the "x" used at the deviation point as the result of applying a deep (unrolled) neural network to the base features (eg word indices). The contextual bandit problem, then, is to learn that neural network well. This basically reduces the question to: are there good CB algorithms for learning neural networks. But the analysis for RESLOPE holds.
>
> In the multi-deviation case, things are much more complicated. In fact, this is one of the things that blocked us from a good analysis in the multi-deviation setting. The problem is that if you deviate at steps 2 and 5, what might be good for improving the reward prediction at step 2 could be bad for step 5 or vice versa, because these two decisions are tied through the network structure as well as the action sequence. (This issue also arises in other learning to search algorithms, like CPI and Searn, which effectively use a sufficiently small learning rate the ensure that there's only one deviation per episode.)
>
>
> Modeling Notions of future Reward
>
> Rather than modeling the Q-function, RESLOPE aims at modeling the advantage function instead, which could be easier to learn in several cases. Learning either the Q-function or the advantage function is sufficient for extracting a greedy policy. Lemma 1 shows that the difference in total loss between two policies can always be computed exactly as a sum of per-time-step advantages of one over the other. We chose to learn the advantages rather than Q-functions as it might be easier to learn and more local. For example, in POS tagging, learning advantages corresponds to learning whether or not the policy made a prediction mistake at a single word which is much easier to learn than the Q-function which requires keeping track of the number of mistakes made from the beginning of the sequence.
>
> Reference Policy Used & Value for Beta
>
> For the structured prediction experiments, the reference policy is a pre-trained model on supervised data (Appendix G). The roll-out probability β is a hyper-parameter that we tune along all the other hyperparameters as described in Appendix H. We pick the best value for β from the set: {0.0, 0.5, 1.0}.
>
> For the reinforcement learning experiments, we don’t assume access to a reference policy and the roll-out probability β is always set to zero.
>
> Note, though, that in the multi-deviation algorithm, there is not a separate notion of a "rollout" policy, like there is in the single-deviation setting.
>
> Difference in performance between the RL and Structured Prediction
>
> This is a good question that unfortunately we don't have a good answer to; we are particularly confused by the poor performance of RESLOPE on cartpole, which is the only place where its behavior is really subpar to even simple approaches like reinforce with baseline (reinforce without a baseline fails quite poorly here, much worse than RESLOPE). This could partially be because RESLOPE came out of a line of work focusing on structured prediction and so the algorithmic style simply is a better fit there, but that's not at all a convincing answer. More work is needed here.

---

> > ### Comment · AnonReviewer3 · 2018-01-28
> > **acknowledgement of author response**
> >
> > This comment acknowledges the author response.  My official review has been edited.

---

> ### Author Response · Authors · 2018-01-05
> **Results for 1-Deviation vs Multiple Deviation**
>
> It’s true that the empirical results for the one-step deviation setting is are worse (particularly in terms of the number of samples needed to learn) than doing multiple deviations. While we don’t have a theoretical analysis for the multi-deviation case, empirically, we found this to be crucial empirically. Although the generated samples for the same episode are not independent, this is made-up for by the huge increase in the number of available samples for training. This is a case where there is a gap between what we can prove theoretically and what works best in practice. We can restructure the outline of the paper to promote the display of the 1-step deviation results on earlier exposure.

---

### Official Review · AnonReviewer1 · 2017-11-27
**Fairly novel approach for solving credit assignment in sparse reward RL, however comparison with other algorithms and explanations arent thorough enough to know if its a significant advance**

**Rating:** 6
**Confidence:** 4

**Review:**

The authors present a new RL algorithm for sparse reward tasks. The work is fairly novel in its approach, combining a learned reward estimator with a contextual bandit algorithm for exploration/exploitation. The paper was mostly clear in its exposition, however some additional information of the motivation for why the said reduction is better than simpler alternatives would help.

Pros
1. The results on bandit structured prediction problems are pretty good
2. The idea of a learnt credit assignment function, and using that to separate credit assignment from the exploration/exploitation tradeoff is good.

Cons:
1. The method seems fairly more complicated than PPO / A2C, yet those methods seem to perform equally well on the RL problems (Figure 2.). It also seems to be designed only for discrete action spaces.
2. Reslope Boltzmann performs much worse than Reslope Bootstrap, thus having a bag of policies helps. However, in the comparison in Figures 2 and 3, the policy gradient methods dont have the advantage of using a bag of policies. A fairer comparison would be to compare with methods that use ensembles of Q-functions. (like this https://arxiv.org/abs/1706.01502 by Chen et al.). The Q learning methods in general would also have better sample efficiency than the policy gradient methods.
3. The method claims to learn an internal representation of a denser reward function for the sparse reward problem, however the experimental analysis of this is pretty limited (Section 5.3). It would be useful to do a more thorough investigation of whether it learnt a good credit assignment function in the games. One way to do this would be to check the qualitative aspects of the function in a well understood game, like Blackjack.

Suggestions:
1. What is the advantage of the method over a simple RL method that predicts a reward at every step (such that the dense rewards add up to match the sparse reward for the episode), and uses this predicted dense reward to perform RL? This, and also a bigger discussion on prior bandit learning methods like LOLS will help under the context for why we’re performing the reduction stated in the paper.

Significance: While the method is novel and interesting, the experimental analysis and the explanations in the paper leave it unclear as to whether its significant compared to prior work.

Revision: I thank the authors for addressing some of my concerns. The comparison with relative gain of bootstrap wrt ensemble of policies still needs more thorough experimentation, but the approach is novel and as the authors point out, does improve continually with better Contextual Bandit algorithms. I update my review to 6.

---

> ### Author Response · Authors · 2018-01-05
> **Authors' Response to AnonReviewer1**
>
> Authors’ Response to highlighted cons:
>
> RESLOPE is more complicated than PPO / A2C
>
> We suppose this depends on how "complicated" is measured. Given a known, fixed contextual bandit algorithm, RESLOPE becomes quite straightforward to implement, certainly more simple (in lines of code) than PPO/A2C would be if you did not have access to, for instance, an autodiff toolkit. Given good lower-level abstractions (autodiff, CB, etc.), both are quote straightforward to implement. Furthermore, RESLOPE comes with significant advantage over PPO/A2C: RESLOPE continually improves as better contextual bandit algorithms become available, a property lacked by PPO/A2C. RESLOPE also fares well empirically, and comes with some nice theoretical guarantees (which, for instance, A2C lacks).
>
>
> RESLOPE and Continuous Action Spaces
>
> It’s true that RESLOPE is designed for discrete action spaces. Extension to continuous action spaces remains an open problem. We have updated the discussion section (section 6) to include this extension as a future work.
>
> Comparison with Ensemble Learning Methods
>
> This is a great question, thank you for raising it! Indeed, the original submission did not adequately separate gains from more complex representation (bag of policies) and alternative estimation methods (bootstrap).
>
> To address this, we have done an addition set of experiments to answer the following question empirically: what is the relative gain of bootstrap exploration with respect to using an ensemble of policies.
>
> Ensemble exploration trains a bag of multiple policies simultaneously. Each policy in the bag generates a Boltzmann probability distribution over actions. These probability distributions are aggregated by averaging. An action is sampled from the aggregated distribution. This has the property of identical network representation, but not using the Bootstrap estimation method.
>
> The result is that in the MTR setting, the Ensemble method is worse than Bootstrap by factors of 3.52, 0.757 and 0.815 respectively on the first three RL tests and, surprisingly, better by a factor of 6.39 on the last. We plan to complete these experiments with more rigor, and extend to the SP setting, in a final version.
>
> Evaluating the learned loss representation for a well-understood RL Environment
>
> We added additional experiments for evaluating the learned loss representation in the grid world reinforcement
> learning environment (Appendix K). This experiment is akin to the experimental setting in section 5.3, however
> it’s performed on the grid world RL environment, where the quantitative aspects of the loss function is well understood. Results are very similar to the structured prediction setting (section 5.3). Performance is better when the loss is additive vs non-additive.
>
> Authors’ Response to proposed suggestions:
>
> RESLOPE & Reward Prediction at Every Step
>
> We’re not aware of a different way for learning the reward in every time step without computing the residual loss as we do in RESLOPE. After estimating the residual losses, RESLOPE reduces the problem to a contextual bandit oracle. This is crucial for accounting for the exploration probability and is necessary for obtaining an unbiased and convergent estimates for the loss. It’s not clear how standard RL can account for the exploration probability when the estimated rewards is used instead of the true reward values, and thus, we didn’t consider this approach in our experiments. (But we're open to suggestions!)
>
> RESLOPE vs LOLS
>
> Both RESLOPE and the bandit version of LOLS (Chang et al., 2015) aim to learn from sparse reward signals by building on the bandit learning to search frameworks. As highlighted in the discussion section (Section 6), they differ significantly in both theory and practice:
>  The “bandit” version of LOLS was analyzed theoretically but not empirically in the original paper; Sharaf & Daumé (2017) found that it failed to learn empirically;
> RESLOPE learns a representation for the episodic loss as a decomposition over time-steps, while LOLS learns directly from the episodic loss signal, this is prone to high variance and doesn’t work in practice (Sharaf & Daumé 2017);
> RESLOPE separates the problem of credit assignment from the exploration problem via a reduction to a contextual bandit oracle. This enables the usage of better variance reduction techniques (e.g. Doubly Robust cost estimation & Multi-task Regression) as well as different exploration algorithms (e.g. bootstrap exploration). LOLS can only use Inverse Propensity Scoring and greedy exploration.

---

> > ### Author Response · Authors · 2018-01-05
> > **References**
> >
> > [1] Kai-Wei Chang, Akshay Krishnamurthy, Alekh Agarwal, Hal Daume ́, III, and John Langford. Learning to search better than your teacher. In Proceedings of the 32Nd International Conference on International Conference on Machine Learning - Volume 37, ICML’15, pp. 2058–2066. JMLR.org, 2015. URL http://dl.acm.org/citation.cfm?id=3045118.3045337.
> > [2] Amr Sharaf and Hal Daume ́, III. Structured prediction via learning to search under bandit feedback. In Proceedings of the 2nd Workshop on Structured Prediction for Natural Language Processing, pp. 17–26, Copenhagen, Denmark, September 2017. Association for Computational Linguistics. URL http://www.aclweb.org/anthology/W17-4304.
> > [3] Miroslav Dud ́ık, Dumitru Erhan, John Langford, and Lihong Li. Doubly robust policy evaluation and optimization. Statist. Sci., 29(4):485–511, 11 2014. doi: 10.1214/14-STS500. URL https: //doi.org/10.1214/14-STS500.

---

### Author Response · Authors · 2018-01-05
**Paper Structure, Exposure, and List of Changes**

The authors appreciate the reviewer’s suggestions for improving the overall exposure of the paper. In order to make it easier for reviewers’ to track the changes we kept the structure largely consistent with the original submission, but we’ll take all of these comments into account in the final version.

@ AnonReviewer3
Thanks for the clarification suggestions on the analysis; we can add explicit definitions of J, Q and V in the background material. The terms in the parentheses are the CB costs because these are exactly the residuals computed and shown as costs to the CB algorithm by construction (essentially the analysis says exactly what these costs should be). We will try to find a way to make this clearer. The issue of non-stationarity is discussed below in greater detail.

List of changes in this version:

1) Extended the discussion sections (section 6) to include some of the open problems and comments highlighted by the reviewers;
2) Added Appendix K. This appendix includes experiments performed for the analysis of the loss representation for the grid world environment;
3) Fixed all the typos highlighted by the reviewers;
4) Updated Appendix H to include the set of values used for tuning the roll-out probability beta.

---

> ### Comment · AnonReviewer3 · 2018-01-28
> **CB cost & analysis**
>
> I'm still confused on this point.  The terms in the (...) in Eq.7 uses the "true" Q values.  Whereas the RESLOPE algorithm uses the CB.cost function (Line 17 & Line 20 in Alg 1).  As far as I can tell, CB.cost is an **estimate** of the (dis)advantage.  Thus, I don't understand how "is exactly the expected value of the target of the contextual bandit cost", as stated in the proof of Theorem 1.  Are you saying that, CB.cost used in Line 20 of Alg 1 is an unbiased estimate?
>
> Everything else about the paper seems OK to me, modulo polishing.

---

> > ### Author Response · Authors · 2018-01-28
> > **CB cost & analysis - Response**
> >
> > Thank you for asking about this; indeed, you're right, there's a missing term. In going from Eq 7 to Eq 8 as it stands right now, we're assuming that we have access to exact quantities, which is not actually the case in practice. In order to account for this, we need to add an additional term \epsilon_{CS} that captures the regret of the cost-sensitive learner. This will then be an additive term in Eq 8. Under a realizability assumption this will go to zero over time, so the impact on Theorem 1 is that an additional realizability assumption is required, or (probably better) to explicitly pull \epsilon_{CS} out as an additional approximation error term in the final bound. We really appreciate you catching this!

---

> > > ### Comment · AnonReviewer3 · 2018-01-28
> > > **Interactions between two online learning reductions?**
> > >
> > > Hmm, I'm not sure the solution is that simple.
> > >
> > > CB.cost is predicting the advantage cost of pi^mix or pi^learn, both of which are evolving functions because the policy is learning over time.  Hence, I don't see a realizability assumption as reasonable except for characterizing the CB.cost of the final pi you learn.
> > >
> > > As for the regret analysis, there is an issue with two interacting online learning reductions, one for learning CB.Cost and one for learning Pi (i.e., CB.Act).  The regret analysis of CB.Act will depend on the convergence of CB.Cost and vice versa.
> > >
> > > This issue arises in other settings:
> > > -- Learnability of (approximate) Nash equilibria in two-player zero-sum games by using two no-regret online learning algorithms for the two players.  The convergence analysis of each player depends on the convergence of the other player.
> > > [1] http://www.cs.cmu.edu/~avrim/ML07/lect1028-1102.pdf
> > >
> > >
> > > -- Convergence analysis in online learning of GANs (where both the generator and discriminator are trained via online learning):
> > > [2] https://arxiv.org/abs/1706.03269
> > >
> > > -- Convergence analysis of sparring-style reductions of the dueling bandits problem:
> > > [3] https://arxiv.org/abs/1502.06362
> > > [4] https://arxiv.org/abs/1705.00253
> > >
> > > In [1] and [3], one resorts to using online learning algorithms with adversarial guarantees (which includes settings where the "environment" is influenced by another online learning algorithm).
> > >
> > > In [2] and [4], the authors are able to more carefully analyze the structure of the interaction, and do not have to resort to the adversarial setting (online learning algorithms with adversarial guarantees can be very inefficient in practice).
> > >
> > > In lieu of more carefully analyzing the interaction between these two online learning procedures, I suspect you'll have to resort to online learning algorithms (for CB.Cost and CB.Act) that have guarantees in the adversarial setting.  I think that will probably work out, but this whole discussion needs to be much more thoroughly fleshed out in the paper.

---

### Decision · Program_Chairs · 2018-01-29
**ICLR 2018 Conference Acceptance Decision**

**Decision:**

Accept (Poster)

**Comment:**

The reviewers agree that the problem of learning learning credit assignment from terminal rewards is interesting, and that the presented approach is promising. There are some concerns regarding the rigor and correctness of the theoretical results, and I ask the authors to improve those aspects of the paper. I also ask the authors to the result figures easier to read. The chosen colors are not ideal and the use of log-scale x-axis is not standard. Finally, including DAgger in the same plot is confusing assuming that DAgger user more information.

---

> ### Author Response · Authors · 2018-05-07
> **Brief thank you and comment on colors**
>
> Sorry for the much-delayed reply. First, thanks for the feedback. We worked hard to address the issues that arose in the theoretical analysis, and are hugely indebted to Reviewer 3 for being helpful here. We appreciate the AC and reviewers for taking so much time for this paper!
>
> One comment on the colors in the plots. The colors are non-standard, but they were chosen specifically to be both color-blind safe and print-friendly, per ColorBrewer2. (Unfortunately we had to add one additional color because there were no color-blind safe "qualitative" colorschemes that contained six colors.) If anyone knows of a better set of colorblind colors we would like to take this into account in future papers, but for now did not change them because they're the best we know of right now.